

# The Southern Annular Mode (SAM) influences phytoplankton communities in the seasonal ice zone of the Southern Ocean

Bruce L. Greaves[1], Andrew T. Davidson[2], Alexander D. Fraser[3,1], John P. McKinlay[2], Andrew Martin[1], Andrew McMinn[1], and Simon W. Wright[1,2]

[1]Institute for Marine and Antarctic Studies, University of Tasmania, Private Bag 129, Hobart, Tasmania 7001, Australia
[2]Australian Antarctic Division, Department of the Environment and Energy, 203 Channel Highway, Kingston, Tasmania 7050, Australia
[3]Antarctic Climate & Ecosystems Cooperative Research Centre (ACE CRC), University of Tasmania, Private Bag 80, Hobart, Tasmania 7001, Australia

**Correspondence:** Bruce Greaves (bruce.on.aria@gmail.com)

**Abstract.** Ozone depletion and climate change are causing the Southern Annular Mode (SAM) to become increasingly positive, driving stronger winds southward in the Southern Ocean (SO), with likely effects on phytoplankton habitat due to changes in ocean mixing, nutrient upwelling, and sea ice. This study examined the effect of the SAM and other environmental variables on the abundance of siliceous and calcareous phytoplankton in the seasonal ice zone (SIZ) of the SO. Samples were collected

5 during repeat transects between Hobart, Australia, and Dumont d'Urville, Antarctica, centred around longitude 142° E, over 11 consecutive austral spring-summers (2002 – 2012). Twenty-two taxa, comprised of species, genera or higher taxonomic groups, were analysed using CAP analysis, cluster analysis and correlation. The SAM significantly affected phytoplankton community composition, with the greatest influence exerted by a SAM index averaged across 57 days centred on 11th March in the preceding autumn, explaining 13.3 % of the variance of taxa composition during the following spring-summer, and

10 showing correlation with the relative abundance of 12 of the 22 taxa resolved. The day through the spring-summer that a sample was collected exerted the greatest influence on phytoplankton community structure (15.4 % of variance explained), reflecting the extreme seasonal variation in the physical environment in the SIZ that drives phytoplankton community succession. The response of different species of *Fragilariopsis* spp. and *Chaetoceros* spp. differed over the spring-summer and with the SAM, indicating the importance of species-level observation in detecting subtle changes in pelagic ecosystems. This study indicated

15 that higher SAM favoured increases in the relative-abundance of large *Chaetoceros* spp. that predominated later in the spring-summer and reductions in small diatom taxa and siliceous and calcareous flagellates that predominated earlier in the spring-summer. Such changes in the taxonomic composition of phytoplankton, the pasture of the SO and principal energy source for Antarctic life, may alter both carbon sequestration and composition of higher tropic levels of the SIZ region of the SO.





## 1 Introduction

Phytoplankton are the primary produces that feed almost all life in the oceans. Seasonal phytoplankton blooms in the seasonal ice zone (SIZ) of the Southern Ocean (SO) feed swarms of krill which, in turn, are key food for sea-birds, fish, whales and almost all Antarctic life (Smetacek, 2008; Cavicchioli et al., 2019). Phytoplankton also play a critical role in ameliorating global climate change by capturing carbon through photosynthesis. Around one third of the carbon fixed by phytoplankton in SIZ of the SO sinks out of the surface ocean (Henson et al., 2015), appreciably more than the global average of around 20 % (Boyd and Trull, 2007; Ciais et al., 2013; Henson et al., 2015). This sequestration of carbon to the deeper ocean is thought to last for climatically significant periods of time, likely hundreds to thousands of years (Lampitt and Antia, 1997). Consequently, SO phytoplankton play a role in mitigating the accumulation of anthropogenic greenhouse gasses in the world's atmosphere (Boyd and Trull, 2007; Deppeler and Davidson, 2017). Any changes in the composition and abundance of phytoplankton in the SIZ are likely to influence both the trophodynamics of the SO and the sequestration of atmospheric carbon.

Global standing stocks of phytoplankton are estimated to be declining at around 1 % per year, a decline largely attributed to rising surface ocean temperature (Boyce et al., 2010). Furthermore, global phytoplankton productivity is predicted to drop by as much as 9 % from years 1990 to 2090 (RCP8.5 *Business As Usual*), with a decline across most of the Earth's ocean area (Bopp et al., 2013). In contrast, higher latitudes, including the SIZ of the SO, are predicted to experience an increase in productivity due to (1) reduced seasonal ice extent and duration leading to the water column receiving more light for longer (Parkinson, 2019; Turner et al., 2013) and/or (2) increased upwelling of nutrient-rich deep ocean water at the Antarctic Divergence (Steinacher et al., 2010; Bopp et al., 2013; Carranza and Gille, 2015).

### 1.1 Importance of the SIZ phytoplankton bloom

The Antarctic SIZ is one of the most productive parts of the SO south of 60˚S (Carranza and Gille, 2015). It is also a significant component of the global carbon cycle by virtue of both carbon sequestration and export by phytoplankton (Henson et al., 2015) as well as upwelling of carbon-rich deep ocean water (Takahashi et al., 2009). It is one of the largest and most variable biomes on Earth, with sea ice extent varying from around 20 million km$^2$ during winter to only 4 million km$^2$ in summer (Turner et al., 2015; Massom and Stammerjohn, 2010). The most macronutrient-rich surface waters of the SIZ occur over the Antarctic Divergence, a circumpolar region of the SO at around 63°S where carbon- and nutrient-rich water upwells to the surface, supplying the nutrients that drive much of the phytoplankton production in the SO (Lovenduski and Gruber, 2005; Carranza and Gille, 2015) and releasing $CO_2$ into the atmosphere (Takahashi et al., 2009).

In winter, phytoplankton growth is limited by light availability and temperature. In spring and summer, phytoplankton can proliferate in the high light, high nutrient waters that trail the southward retreat of sea ice (Wilson et al., 1986; Smetacek and Nicol, 2005; Lannuzel et al., 2007; Saenz and Arrigo, 2014; Rigual-Hernández et al., 2015). The SIZ supports high phytoplankton standing stocks and productivity in waters where phytoplankton abundance in blooms can double every few days (Wilson et al., 1986; Sarthou et al., 2005). Phytoplankton productivity in the SIZ is generally highest around the time of





maximum solar irradiation (Smith and Asper, 2001) but is characterised by large-scale spatial and temporal variability (Martin et al., 2012) with only 17-24 % of ice edge waters experiencing phytoplankton blooms in any spring-summer period. Wind speed is the primary determinant of phytoplankton bloom development in the SIZ, with calmer conditions fostering shallow mixed depths that maintain phytoplankton cells in a high light environment and maximise productivity (Savidge et al., 1996; Fitch and Moore, 2007).

## 1.2    The Southern Annular Mode

The Southern Annular Mode (SAM), which is also variously also called the High-Latitude Mode and the Antarctic Oscillation, has been defined as the difference in normalised zonal mean sea-level pressure between 40°S and 65°S (Gong and Wang, 1999; Marshall, 2003). The SAM is the principal mode of atmospheric circulation at high latitudes of the Southern Hemisphere, and variation in the SAM typically describes around 35 % of total Southern Hemisphere climate variability (Marshall, 2007). The SAM is currently the dominant large-scale mode through which climate change is expressed on the SO (Thompson and Solomon, 2002; Lenton and Matear, 2007; Lovenduski, 2007; Swart et al., 2015).

There is a trend toward increasing SAM from 1979 to 2017 of 0.011 index points per year (NOAA, 2017), attributed to both ozone-depletion (Thompson and Solomon, 2002; Arblaster and Meehl, 2006; Gillett and Fyfe, 2013; Jones et al., 2016) and to increasing atmospheric greenhouse gas concentrations (Thompson et al., 2011). The long-term average SAM index is now at its highest level for at least the past 1,000 years (Abram et al., 2014). Continuing increases in atmospheric greenhouse gasses are expected to drive further increase in the SAM index in all seasons (Arblaster and Meehl, 2006; Swart and Fyfe, 2012; Gillett and Fyfe, 2013), despite the expected recovery in stratospheric ozone concentrations to pre-ozone hole values by around 2065 (Son et al., 2009; Schiermeier, 2009; Thompson et al., 2011; Solomon et al., 2016).

More positive SAM has been associated with lower atmospheric pressure at sea level and increased storminess (Kwok and Comiso, 2002; Hall and Visbeck, 2002; Marshall, 2007). These changes are particularly marked south of 60°S in the atmospheric Southern Circumpolar Trough (Hines et al., 2000; Mackintosh et al., 2017), a region characterised by strong winds with variable direction (Taljaard, 1967). Stronger winds may result in increased transport of surface water northward from the Antarctic Divergence by Ekman drift (Lovenduski and Gruber, 2005; DiFiore et al., 2006), potentially driving increased upwelling of nutrient- and carbon-rich deep ocean water at the Antarctic Divergence (Hall and Visbeck, 2002). More positive SAM is also associated with reduced near-surface air temperature over the SIZ due to an increased frequency of strong southerly winds and increased cloud cover (Lefebvre et al., 2004; Sen Gupta and England, 2006). Sea ice extent shows zonal relationships with the SAM: positive relationships between the SAM and sea-ice extent in the Western Pacific and Indian sectors of the SO (the Indian sector was sampled in this work) and negative or non-existent relationships in other sectors (Kohyama and Hartmann, 2016). Wind also affects the nature of the sea ice, breaking up floes, increasing flooding, and if blowing from the south, both opening the pack ice and leading to new frazzle ice formation (Massom and Stammerjohn, 2010). Lower sea-surface temperatures have been observed to lag positive SAM events by one to four months (Lefebvre et al., 2004; Meredith et al., 2008). Such changes in the SAM may take weeks to months to be manifested in phytoplankton communities (Sen Gupta





and England, 2006; Meredith et al., 2008), while extreme SAM events might impact phytoplankton community composition for multiple years (Ottersen et al., 2001).

By mediating upwelling, ocean mixed depth, air temperature, and sea-ice characteristics and duration, it is likely that increases in the SAM will affect the composition and abundance of phytoplankton in the SIZ of the SO. Lovenduski and Gruber (2005) predicted that increased SAM would support higher phytoplankton productivity, and subsequent analyses by Arrigo et

al. (2008); Boyce et al. (2010), and Soppa et al. (2016) have confirmed a positive relationship between the SAM and phytoplankton standing stocks and productivity south of 60°S in the SIZ.

### 1.3 The Hypothesis

Based on the predicted and observed positive relationships between the SAM and phytoplankton productivity and biomass in the SIZ of the SO, we hypothesised that changes in the SAM could also elicit changes in the composition and abundance of the

phytoplankton community. To test this hypothesis, we conducted a scanning electron microscopic survey of hard-shelled phytoplankton in surface waters of the Antarctic SIZ using samples collected between October and February each spring-summer over 11 consecutive years (2002/03 – 2012/13). We then related the composition of these communities to environmental variables including the SAM.

## 2 METHODS

Fifty-two surface-water samples were collected from the seasonal ice zone (SIZ) of the Southern Ocean (SO) across 11 consecutive austral spring-summers from 2002/03 to 2012/13. The samples were collected aboard the French re-supply vessel MV L'Astrolabe during resupply voyages between Hobart, Australia, and Dumont d'Urville, Antarctica, between the 20th October and the 1st March. Most samples were collected from ice-free water, although some were collected south of the receding ice-edge (Fig. 1a).

The sampled area was in the high latitude SO (Fig. 1b) in the south-east corner of the Australian Antarctic Basin, spanning 270 km of latitude between 62°S and 64.5°S, and 625km of longitude between 136°E and 148°E. The area lies >100 km north of the Antarctic continental shelf, in waters >3,000 m depth.

Samples were obtained from the clean seawater line of the re-supply ship from around 3 m depth. Each sample represented 250 ml of seawater filtered through a 25 mm diameter polycarbonate-membrane filter with 0.8 $\mu$m pores (Poretics). The filter

was then rinsed with two additions of approximately 2 ml of MilliQ water to remove salt, then air dried and stored in a sealed container containing silica gel desiccant. Samples were prepared for scanning electron microscope (SEM) survey by mounting each filter onto metal stubs and sputter coating with 15 nm gold or platinum. Only organisms possessing hard siliceous or calcareous shells were sufficiently well preserved through the sample preparation technique that they could be identified by SEM, and included diatoms, coccolithophores, silicoflagellates, Pterosperma, parmales, radiolarians, and armoured dinoflagellates.





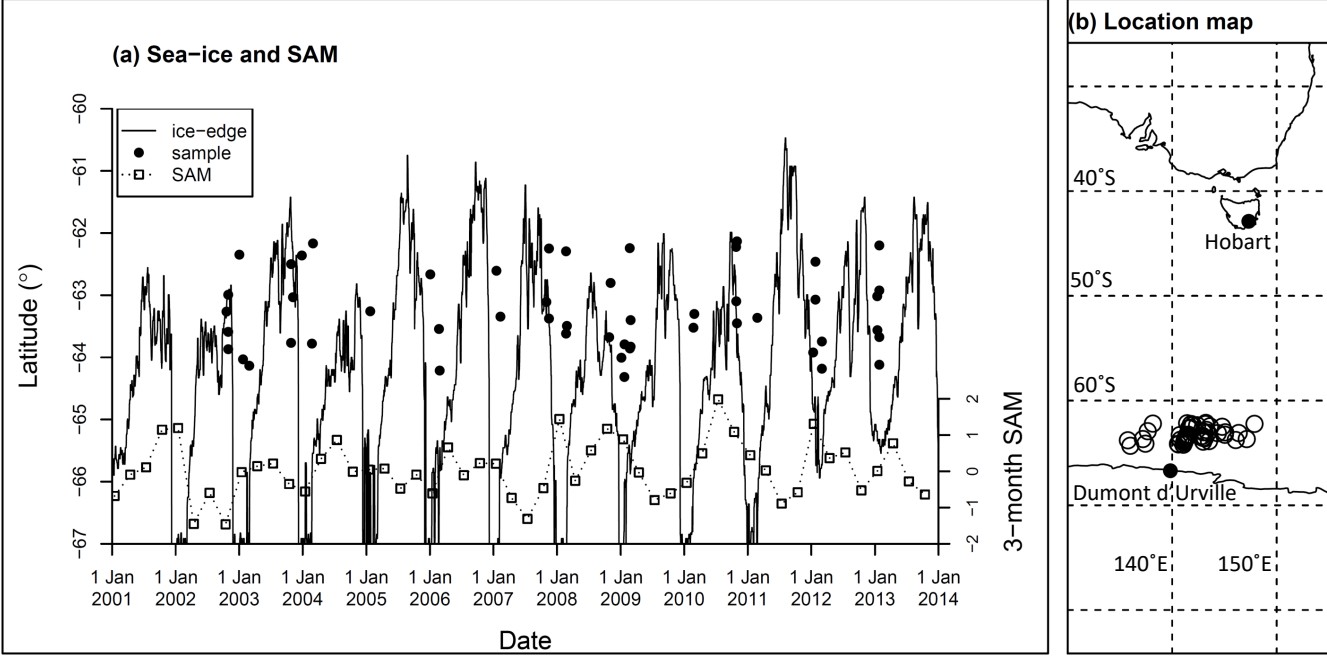

**Figure 1.** (a) Latitude and timing of samples (filled circles), sea ice extent at 143˚E (solid line), and three-month season-average values of daily SAM (open squares). Latitude -67˚ denotes the Antarctic coastline. (b) Sampling area in relation to southern Australia and the Antarctic coastline, with sample locations indicated as open circles.

## 2.1 Phytoplankton composition and abundance

The composition and abundance the phytoplankton community of each sample was determined with the aid of a JEOL JSM 840 Field Emission SEM. Cell numbers for each phytoplankton taxon were counted in randomly selected digital images of SEM fields taken at x400 magnification (Fig. 2). Each image represented an area of 301 x 227 $\mu$m (0.068 mm$^2$) of each sample filter, which was captured at a resolution 8.5 pixels per $\mu$m. A minimum of three SEM fields were assessed for each sample, with more fields assessed when cell densities were lower. On average, 387 cells were counted for each sample. Taxa were classified with the aid of Scott and Marchant (2005), Tomas (1997), and expert opinion. Cell counts per image were converted to volume-specific abundances (cells per ml) by dividing by 0.0348 ml of sea-water represented by each image.

A total of 19,943 phytoplankton organisms were identified and counted: 18,872 diatoms, 322 Parmales, 173 coccolithophores, 81 silicoflagellates, and 45 Petasaria. A total of 48 phytoplankton taxa were identified, many to species level. Because the diatoms *Fragilariopsis curta* and *F. cylindrus* could not be reliably discriminated at the microscope resolution employed, they were pooled into a single taxa-group. Other taxa were also grouped, namely *Nitzschia acicularis* with *N. decipiens* to a single group, and discoid centric diatoms of the genera *Thalassiosira*, *Actinocyclus* and *Porosira* to another. Rare species, with max-





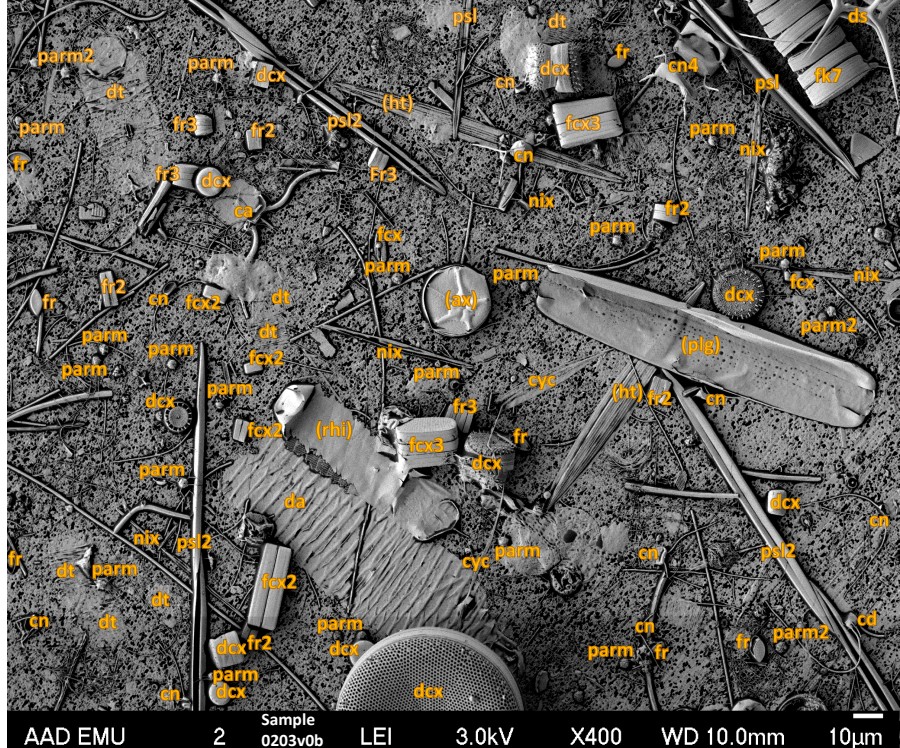

| code | taxa |
|------|------|
| ca | *Chaetoceros atlanticus* |
| cc | *Chaetoceros concavicornis* |
| cca | *Chaetoceros castracanei* |
| cd | *Chaetoceros dichaeta* |
| cn | *Chaetoceros neglectus* |
| coc | *Emiliania huxleyi* (haptophyte) |
| cyc | *Cylindrotheca closterium* |
| da | *Dactyliosolen antarcticus* |
| dcx | discoid centric diatoms |
| ds | *Dictyocha speculum* (silicoflagellate) |
| dt | *Dactyliosolen tenuijunctus* |
| fcx | *Fragilariopsis cylindrus/curta* |
| fk | *Fragilariopsis kerguelensis* |
| fps | *Fragilariopsis pseudonana* |
| fr | *Fragilariopsis rhombica* |
| fri | *Fragilariopsis ritscheri* |
| guc | *Guinardia cylindrus* |
| nix | *Nitzschia acicularis* |
| parm | *Parmales* spp. |
| pet | *Petasaria heterolepis* (other) |
| psl | *Pseudo-nitzschia lineola* |
| ta | *Thalassiothrix antarctica* |

**Figure 2.** Example of phytoplankton identification on a single SEM image. Overlying letters are taxa-codes for individual phytoplankton taxa used in analysis; codes in parenthesis are rare taxa.

imum relative abundance <2 %, were removed from the data prior to analysis as they were not considered to be sufficiently abundant to warrant further analysis (Webb and Bryson, 1972; Taylor and Sjunneskog, 2002; Świło et al., 2016). After pool-

ing taxa and deleting rare taxa, twenty-two taxa and taxonomic-groups (species, groups of species and families) remained to describe the composition of the phytoplankton community.

## 2.2   Environmental covariates

Phytoplankton abundances were related to a range of environmental covariates available at the time of sampling. These included the SAM, sea surface temperature (**SST**), **Salinity**, time since sea ice cover (**DaysSinceSeaIce**, defined below), minimum

latitude of sea ice in the preceding winter, latitude and longitude of sample collection, the days since 1st October that a sample was collected (**DaysAfter1Oct**), the year of sampling (year, being the year that each spring-summer sampling season began), the time of day that a sample was collected, and macro-nutrient concentrations: phosphate ($PO_4$), silicate ($SiO_4$) and nitrate + nitrite (hereafter nitrate, $NO_x$).



Water samples for dissolved macro-nutrients were collected, frozen on ship, and later analysed at CSIRO in Hobart using
standard spectrophotometric methods (Hydes et al., 2010). Daily estimates of SAM were obtained from the US NWS Climate Prediction Center's website and are the NOAA Antarctic Oscillation Index values based on 700-hPa geopotential height anomalies (NOAA, 2017). The variable ***DaysSinceSeaIce*** was defined as the time since sea ice had melted to 20 % cover, after Wright et al. (2010), as determined from daily Special Sensor Microwave/Imager (SSM/I) sea ice concentration data distributed by the University of Hamburg (Spreen et al., 2008).

To examine the lag in the expression of the SAM on phytoplankton community composition, two response surfaces were constructed relating the variance in phytoplankton community composition explained by the SAM to the temporal positioning of the period over which daily SAM was averaged. These were derived by evaluating separate CAP analyses (described below) based on daily SAM averaged across a range of days {1, 3, 5, . . . 365} centred on (i) each calendar day individually (1 Jan – 31 Dec) through the year associated with each sample; and (ii) lagged from 1 to 365 days prior to each sample collection date.

**2.3    Statistical analysis**

Clustering techniques were used to explore similarities in phytoplankton community composition and abundance among samples, and distance-based redundancy analysis (Legendre and Anderson, 1999) and correlation analysis were used to relate community structure to environmental covariates. The abundance data were converted to relative abundance by dividing each abundance estimate by the total abundance of the 22 taxa in the sample, then square-root-transformed to reduce possible dom-
inance of the analysis by a few abundant taxa. Relative abundance was used to alleviate variation among samples as a result of dilution, a phenomenon whereby the abundance of cells can be reduced in a matter of hours by an abrupt increase in wind speed and associated increase in the mixed layer depth (Carranza and Gille, 2015), diluting near-surface cells into a greater water volume. However, relative abundance has the disadvantage that blooming of one species will cause a reduction in relative abundance of other present species, when their absolute abundances may not have changed. The Bray-Curtis dissimilarity
index (Bray and Curtis, 1957) was used to calculate the resemblance of samples based on their community structure. The advantage of this index for the cell count data was that similarity among samples was not strongly affected by the absence of taxa. Hierarchical agglomerative clustering based on average linkage was performed on the Bray-Curtis resemblance matrix. Significant differences among sample clusters were determined according to the similarity profile (SIMPROF) permutation method of Clarke et al. (2008), based on alpha = 0.05 and 1,000 permutations.

Constrained analysis of principal coordinates (CAP, (Anderson and Willis, 2003)) was used to estimate the influence of environmental covariates in explaining community composition. This procedure used the Bray-Curtis resemblance matrix to partition total variance in community composition into unconstrained and constrained components, with the latter representing the variation due to the environmental covariates. A forward selection strategy was used to choose the optimum model containing the minimum subset of constraints required to explain the most variation in phytoplankton community structure
(Legendre et al., 2011). Linear projections of significant covariates were plotted as arrows in the ordination diagram, indicating the direction and magnitude of effects that were correlated with changes in the phytoplankton community (Davidson et al.,



2016). Taxa were added to the CAP plots as weighted site-averages for each species, thereby indicating the relative influence of the fitted environmental constraints on each phytoplankton taxon/group.

Pair-wise correlation analyses were performed using Pearson's correlation coefficient $r$ to explore the relationships amongst
environmental variables, and between these environmental variables and the relative abundances of phytoplankton taxa (Rodgers and Nicewander, 1988). Given the large number of pair-wise correlations considered, we applied a Bonferroni correction to give consideration to family-wise error rate by setting alpha, which is usually $\alpha=0.05$ (Gibbons and Pratt, 1975; Cohen, 1990), to $\alpha/m$ where m is the total number of correlations considered. Recognising that $\alpha/m$ may be conservative (Nakagawa, 2004), we indicated when calculated correlations were significant at both p<0.05 and at Bonferroni corrected p<0.05/m.

Data management and manipulation, summary statistics, correlation analysis, and scatter plots were undertaken in Microsoft Excel (2016) and R (R Core Team, 2016). Cluster analysis and SIMPROF were undertaken using the R package clustsig (Whitaker and Christman, 2014). CAP analyses were conducted using the capscale function in the R package vegan (Oksanen et al., 2015).

## 3   RESULTS

### 3.1   Observed abundance


Abundance of individual taxa averaged 133 cells per ml and ranged to a maximum of 8,796 cells per ml. Of the 22 taxa/groups identified in this study (hereafter taxa), four taxa were identified in all 52 samples and 11 taxa were identified in more than 90 % of samples.

### 3.2   CAP analysis and pair-wise relationships

Empirical identification of the time between variation in the SAM and the manifestation of this variation in the phytoplankton community structure revealed three modes (maxima) in phytoplankton community composition explained by the SAM. The first was an autumn seasonal SAM mode, which was determined to be the average of 57 daily SAM estimates centred on the preceding 11th March (11th Feb – 8th Apr). This mode explained up to 13.3 % of the variance in taxonomic composition (**SAM autumn**, Fig. 3a, Table 1a). The second was a spring seasonal mode, which was determined to be the average of 75 daily
SAM estimates centred on 25th October (20th Sep – 3rd Dec). This mode explained up to 10.3 % of variance in taxonomic composition (**SAM spring**, Fig. 3a, Table 1a). Unlike the other modes that were related to the time of year, the third mode was timed relative to the date of sample collection for each sample and comprised the average of the 97 daily SAM estimates centred 102 days prior to each sample collection date. It explained 9.9 % of the variance in phytoplankton composition (**SAM prior**, Fig. 3b, Table 1a). The mean standard error on estimates of the SAM indices were 0.14 SAM index units for **SAM**
**autumn** and **SAM spring**, and 0.13 for **SAM prior** (Table 2a). Note that **SAM prior** and **SAM spring** temporally overlapped to





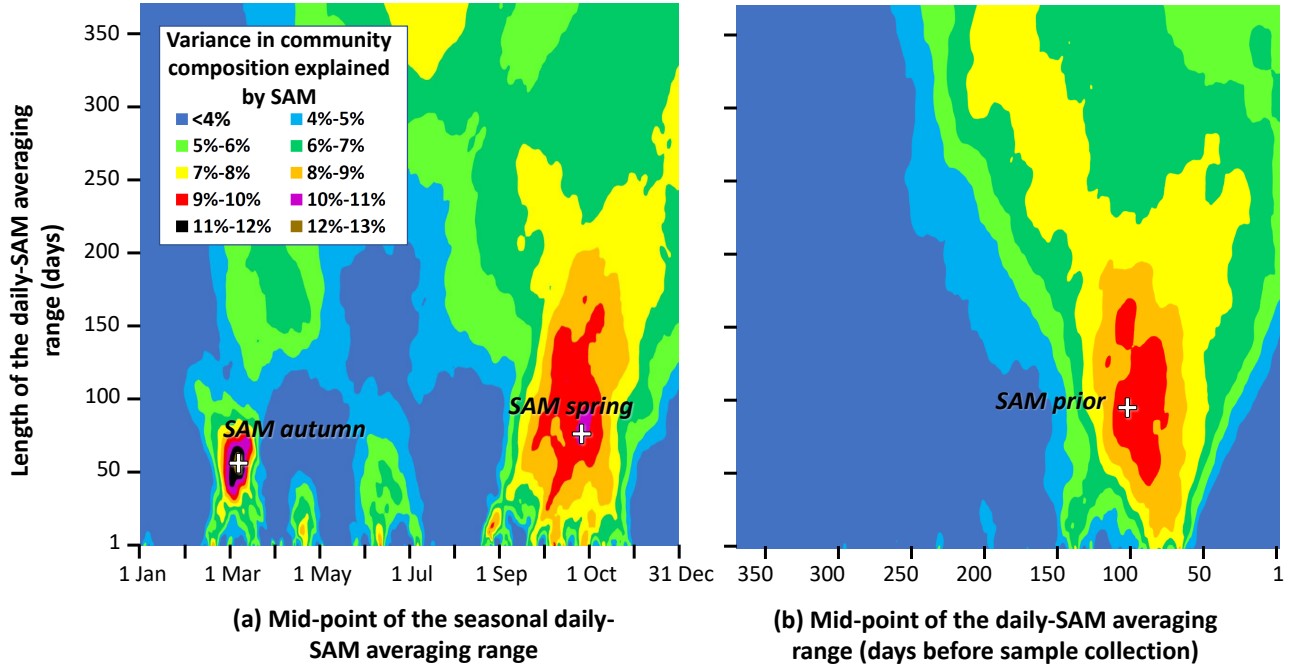

**Figure 3.** Variance in phytoplankton community composition explained by the SAM versus timing and length of the SAM period. Response surfaces relate the fraction of total variance (inertia) in community composition attributable to the SAM, versus the number of days of daily-SAM averaged (vertical axis) and the timing of the range of averaged daily-SAM (horizontal axis). The x-axis is expressed as: (a) the time through the calendar year of the middle of the range; and (b) the number of days before a sample was collected, to the middle of the averaged daily-SAM range. Three obvious modes are identified with crosses (***SAM autumn***, ***SAM spring*** and ***SAM prior***).

varying extents across the 52 samples (Fig. 4) and so were not entirely independent covariates: for example, a sample collected in the summer had previous days contributing to both ***SAM prior*** and ***SAM spring***.

The optimum multi-covariate CAP analysis showed that the autumn mode (***SAM autumn***) explained the most variance in community composition of the three identified SAM modes (12.6 %), while the prior-to-sampling mode (***SAM prior***)

explained a further 4.3 % of variation when fitted as the second constraining SAM covariate (Table 1b). These two SAM indices were moderately and significantly positively correlated (r: 0.51, Table 2c). Both showed similar negative correlations (Table 2b) with the abundances of the small diatoms *Fragilariopsis rhombica* (relationship with ***SAM autumn*** depicted in Fig. 5a) and *Nitzschia acicularis/decipiens*, and the coccolithophorid *Emiliana huxleyi*, and similar positive correlations with the abundances of larger diatoms *Chaetoceros atlanticus*, *Chaetoceros dichaeta* and *Dactyliosolen antarcticus*. A further six

taxa showed a correlation with ***SAM autumn*** but not ***SAM prior***, namely positive correlations with *Chaetoceros concavicornis/curvatus*, *Fragilariopsis kerguelensis* (relationsip with ***SAM autumn*** depicted in Fig. 5b), *Pseudo-nitzschia lineola*, and




**Table 1.** Variance in the relative abundance composition of 22 phytoplankton taxa attributable to constraining environmental covariables in the CAP analysis.

| CAP analysis | variance category | covariate | variance | fraction of total variance | p |
|---|---|---|---|---|---|
| (a) Variables fit individually as the only constraining covariate | | *DaysAfter1Oct* | 0.61 | 15.4 % | <0.001 |
| | | *SST* | 0.57 | 14.6 % | <0.001 |
| | | *SAM autumn* | 0.52 | 13.3 % | <0.001 |
| | | *Long.E* | 0.47 | 11.9 % | <0.001 |
| | | *SAM spring* | 0.41 | 10.3 % | <0.001 |
| | | *SAM prior* | 0.39 | 9.9 % | <0.001 |
| | | *DaysSinceSeaIce* | 0.23 | 5.9% | 0.004 |
| | | *Salinity* | 0.18 | 4.7 % | 0.018 |
| | | *Year* | 0.13 | 3.4 % | 0.086 |
| | | *Long.E* | 0.10 | 2.5 % | 0.228 |
| | | Minimum latitude of sea-ice the previous winter | 0.06 | 1.6 % | 0.537 |
| (b) Optimum multi-covariate model | variance explained by all constraining covariables | | 1.48 | 37.5 % | <0.001 |
| | individual constraining covariables | *DaysAfter1Oct* | 0.61 | 15.4 % | <0.001 |
| | | *SAM autumn* | 0.50 | 12.6 % | <0.001 |
| | | *Long.E* | 0.21 | 5.2 % | <0.001 |
| | | *SAM prior* | 0.17 | 4.3 % | 0.006 |
| | Unexplained residual | | 2.46 | 62.5 % | |
| | Total variance in taxa-composition between samples | | 3.94 | 100 % | |

*Thalassiothrix antarctica*, and negative correlations with *Dactyliosolen tenuijunctus* and the *Parmales*. Three taxa showed correlations with **SAM prior** but not **SAM autumn**, namely positive correlations with *Chaetoceros neglectus* and the silicoflag-ellate *Dictyocha speculum*, and a negative correlation with *Petasaria heterolepis*. In the optimum multi-covariate CAP analysis

(Table 1b, Fig. 6a) the first four CAP axes were statistically significant (p<0.05), the first two axes together explained a total of 31.1 % of the variation in community taxonomic composition, and the third and fourth CAP axis together explained a further 6.4 % (not tabulated).

Following cluster analysis, SIMPROF identified seven significantly different groups (p<0.05), with samples loosely grouped on the basis of their within-season successional maturity (**DaysAfter1Oct**) and the SAM index (Fig. 6b). The coloured groups of

samples in the 2D representation of the optimum multi-covariate CAP analysis (Fig. 6a) are coloured according to the clusters identified in Fig. 6b, with their positioning further indicating the influences of **DaysAfter1Oct** and the SAM index on cluster groupings. This showed samples in clusters 3 and 4 (Fig. 6b) were commonly associated with more positive SAM, while those in clusters 5, 6 and 7 were associated with negative SAM values. Samples in clusters 2 and 5 were commonly collected earlier





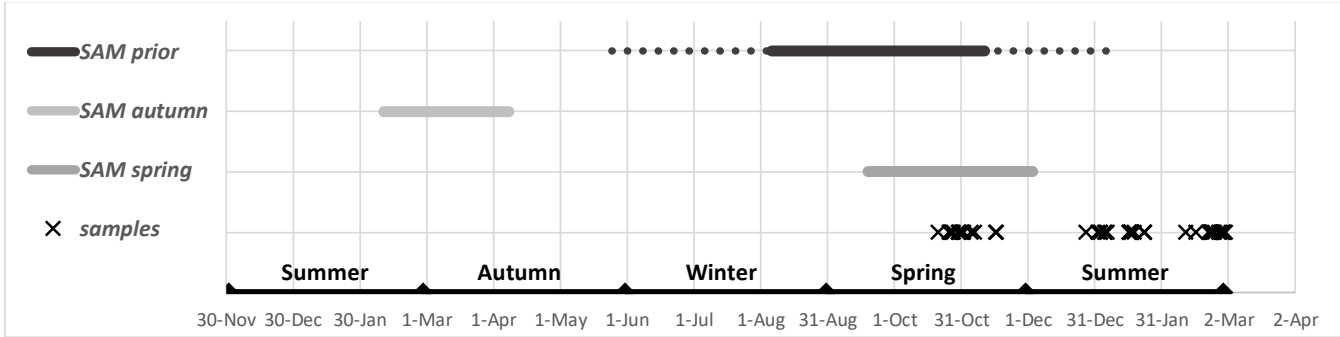

**Figure 4.** Modes of SAM influence on phytoplankton community composition. *SAM prior* was determined relative to sample collection: the depicted solid line represents the average temporal location of the 161-day period and the broken lines represent the earliest and latest extent of the range associated with the earliest and latest samples.

in the spring-summer period (lower *DaysAfter1Oct*) while those in clusters 1, 4, 6 and 7 were commonly collected later (Fig.
225   6).

Fifteen of the 22 taxa showed significant (p<0.05) pairwise correlations with one or more of the SAM modes, with *SAM autumn* being the most influential (Table 2b). Of the 12 taxa showing a correlation between their relative abundance and *SAM autumn* (Table 2b), six also showed a significant correlation with the sample collection date (*DaysAfter1Oct*). Of these, three taxa were negatively correlated with both *SAM autumn* and *DaysAfter1Oct* (i.e. had maximum abundance early in the
season). Conversely, two taxa were positively correlated with both *SAM autumn* and *DaysAfter1Oct*. A similar but stronger relationship was seen between individual taxon correlations with *SAM autumn* and *DaysSinceSeaIce*. That is, taxa showing a negative correlation between relative abundance and *SAM autumn* were more likely to show a negative abundance-correlation with *DaysSinceSeaIce*, i.e those whose maximum relative abundance occurred earlier after the opening of the winter's sea ice, and vice versa (r: 0.49, p<0.05, Fig. 5c). Individual taxon abundance relationships with *SAM spring* and *SAM prior* did not
exhibit trends with individual taxon relationships with either *DaysAfter1Oct* or *DaysSinceSeaIce*.

*SAM prior* and *SAM spring* represented a similar time span in the spring immediately prior to sampling (Fig. 4) and were strongly and significantly correlated (r: 0.83, Table 2c). Samples were collected over a calendar range of 140 days (20 Oct. - 1 Mar., Table 2a) and thus the 163-day period represented by *SAM prior* varied in its position in the calendar across the 140-day spread of the 52 samples (Fig. 4). The modes also showed similar correlation sign with taxonomic abundances (Table 2b).
It was not possible, however, to determine whether the pre-season SAM influence was a spring effect or a prior-to-sampling effect, and whilst both appear to be important explanatory terms, only *SAM prior* was retained in the optimum CAP model (Table 1b).

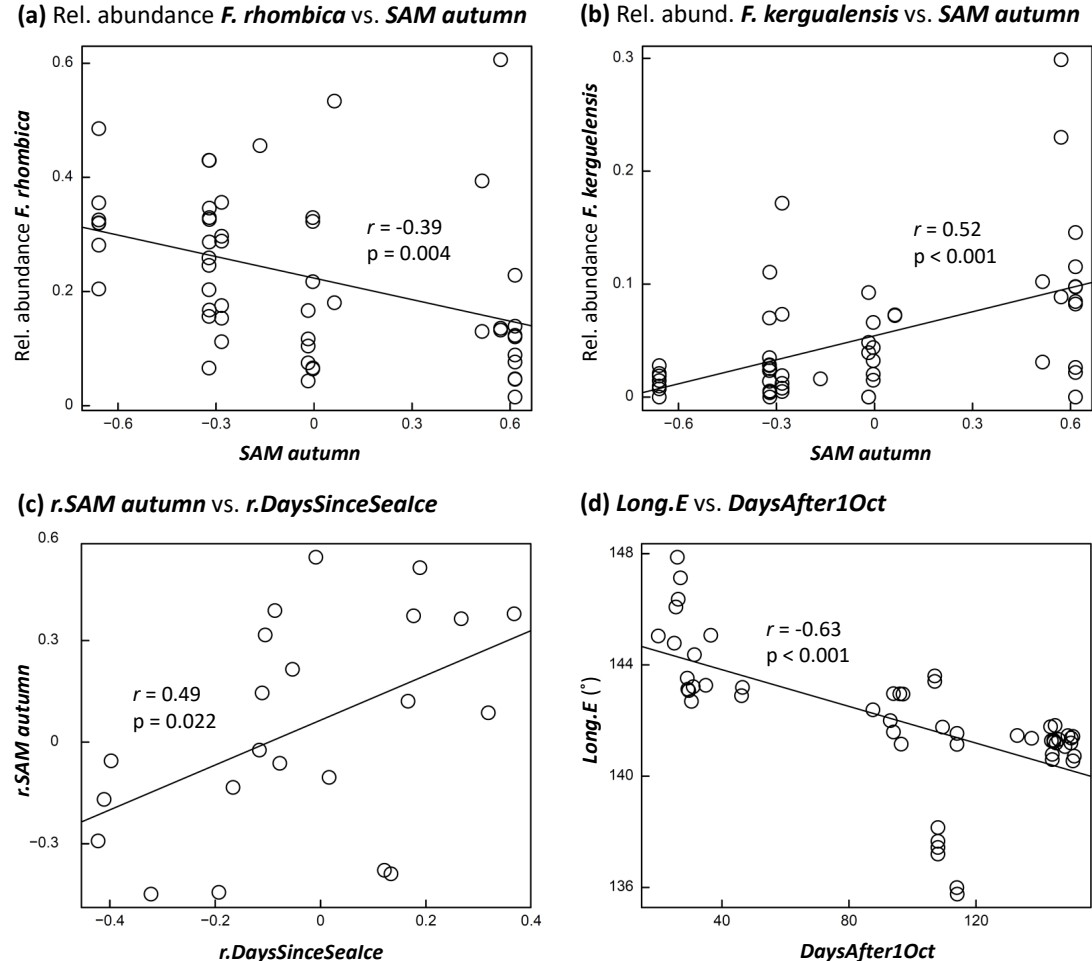

**Figure 5.** Scatter-plots: (a,b) examples of phytoplankton taxon relative abundance versus ***SAM autumn***; (c) taxa correlations with ***SAM autumn*** (*r.SAM autumn*) versus taxa correlations with ***DaysSinceSeaIce*** (*r.DaysSinceSeaIce*); and (d): ***Long.E*** of sample collection versus ***DaysAfter1Oct***. Each figure shows Pearson's correlation coefficient (*r*) and p associated with the relationship. A line of least-squares best fit is provided to give an indication of trend, though clearly several underlying assumptions of linear regression would not be met.

In the optimum multi-covariate CAP model, ***DaysAfter1Oct*** explained the greatest proportion of the observed variance in phytoplankton community composition (Table 1b). This variable captured the seasonal succession of the phytoplankton com-
munity. Alone, it explained up to 15.4 % of the total variation (Table 1b) and its effect on the phytoplankton community in the first two fitted CAP axes was approximately orthogonal to that of the SAM (Fig. 6a). A weak positive relationship was detected between ***SAM autumn*** and ***DaysAfter1Oct*** indicating a weak trend of sampling later in the spring-summer period in years with higher autumn SAM (r: 0.32, Table 2c), but otherwise the SAM indices and ***DaysAfter1Oct*** were unrelated. Ten taxa showed





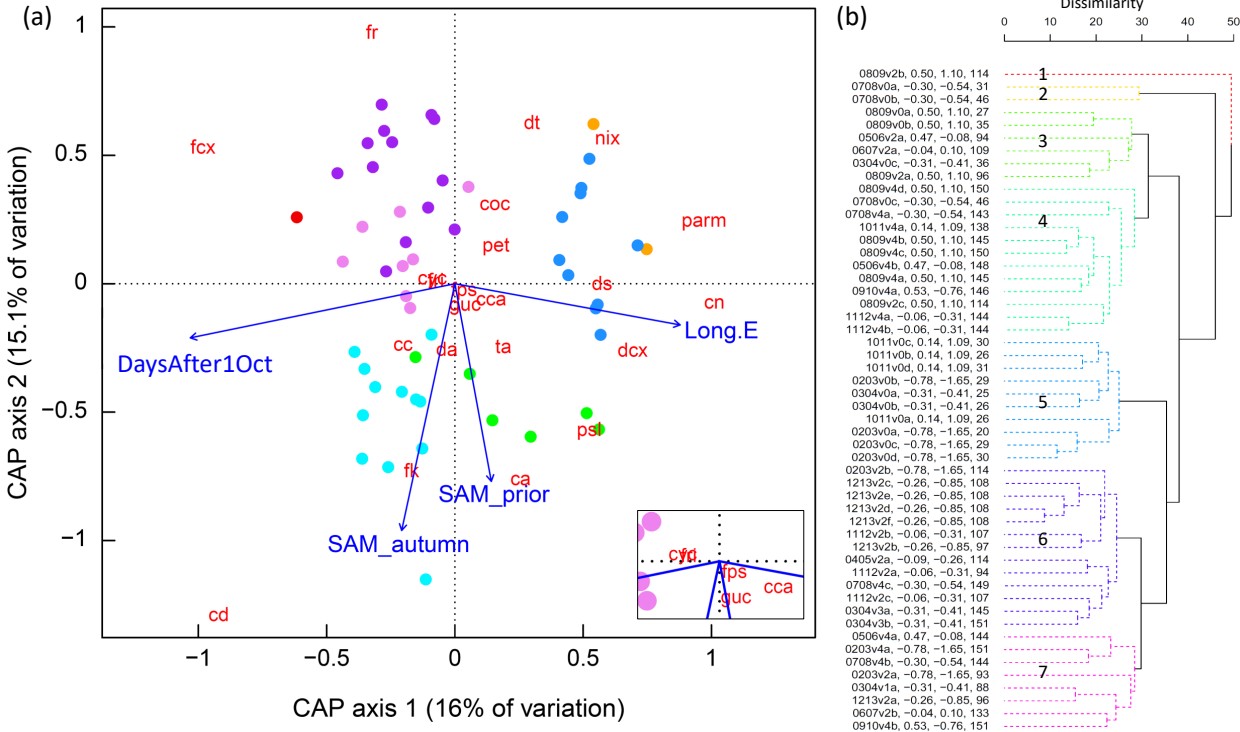

**Figure 6.** (a) CAP analysis of phytoplankton taxonomic composition. Dots represent individual samples, with colours corresponding to significant clusters (Fig. 6b). The 22 phytoplankton taxa/groups are overlain as weighted averages of their sample scores (red abbreviations, after Fig. 2) with positions plotted with a three-times exaggeration of distance from the origin to more easily visualise their relationships with constraining environmental variables. Linear projections of the significant constraining environmental covariates appear as blue arrows, the length and angle of which represents the magnitude and direction of influence of each variable on community composition. The inset shows the taxa located close to the origin, diatoms fri and cyc collocating. (b) Cluster analysis dendrogram of the 52 samples based on similarities in phytoplankton community structure, using colour to show 7 significantly different groups (numbered 1-7, solid lines, $\alpha < 0.05$). Sample labels contain: season and voyage (e.g. 0809v2b = Austral Spring-Summer over 2008-09, voyage designation 2, sample b is the second sample obtained from the SIZ during that voyage); ***SAM autumn*** value, ***SAM prior*** value, and the ***DaysAfter1Oct*** value.

significant (p<0.05) correlation between their relative abundance and ***DaysAfter1Oct*** (Table 2b): *Chaetoceros castracanei*, *C.*
*neglectus*, *D. speculum*, *E. huxleyi*, *N. acicularis/decipiens*, *Parmales*, *P. lineola*, and the discoid centric diatoms showed negative abundance-correlations with ***DaysAfter1Oct*** indicating greatest relative abundance early in the spring-summer, while *C. concavicornis/curvatus* and *C. dichaeta* had greater relative abundance later in the period.

Other environmental covariates that did not significantly influence taxonomic composition were the time though the day that a sample was collected, and the minimum latitude reached by sea ice cover in the previous winter (Extra material S1).



## 3.3 Correlations among taxa

The relative abundances of the 22 phytoplankton taxa were largely unrelated among samples. Of the 231 pairwise correlations between these taxa, only 35 were significantly positive and 18 were significantly negative (p<0.05, Extra material S2). Applying a Bonferroni correction reduced these significant correlations to 15 positive and 8 negative relationships.

## 4 Other relationships

The observed concentrations of the macro nutrients nitrate $NO_x$, phosphate $PO_4$, and silicate $SiO_4$ showed significant negative correlations with ***SAM autumn*** (r= -0.39, -0.56, -0.42 respectively, Table 2d). The concentrations of these nutrients showed stronger negative correlations with ***DaysAfter1Oct*** (r: -0.77, -0.73, -0.56, Table 2d). Macronutrient concentrations were unrelated to either ***SAM prior*** or ***SAM spring*** (Table 2d).

Sea surface temperature (***SST***) and ***DaysSinceSeaIce*** also showed positive correlations with ***DaysAfter1Oct*** (r: 0.92 and 0.56 respectively, Table 2c), and ***Salinity*** and ***Long.E*** showed negative correlations with ***DaysAfter1Oct*** (r: -0.43 and -0.63, Table 2c). When individually fitted as the first constraining covariate in a CAP model, ***SST***, ***Long.E***, ***DaysSinceSeaIce*** and ***Salinity*** explained 14.6 %, 11.9 %, 5.9 % and 4.7 % of variation in phytoplankton community composition respectively (Table 1a). ***SST*** and ***DaysSinceSeaIce*** also showed pairwise relationships with taxa abundances like those detected with ***DaysAfter1Oct***, ***Long.E*** and ***Salinity*** but with opposite correlation sign (Table 2c). Whilst ***SST***, ***DaysSinceSeaIce*** and ***Salinity*** varied systematically through the season, they didn't explain more variance than ***DaysAfter1Oct*** and thus didn't appear in the optimum multi-covariate CAP model (Table 1b). The significant negative correlation between the ***DaysAfter1Oct*** that a sample was collected and the longitude that it was collected (r -0.63, Fig. 5d) indicated that samples collected earlier in the spring-summer were more likely to have been collected further to the east.

Neither relative taxonomic total cell volume, estimated using the method of Hillebrand et al. (1999), or inferred relative taxonomic total cell biomass, estimated using the method of Menden-Deuer and Lessard (2001), showed influence of any of the SAM indices (results not shown).

## 5 DISCUSSION

### 5.1 SAM and phytoplankton community composition

Our results show that the Southern Annular Mode (SAM) does indeed affect the composition and abundance of phytoplankton in the seasonal ice zone (SIZ) of the Southern Ocean (SO), supporting our hypothesis. This conclusion was supported by a combination of three analyses. (i) Permutation-based analyses of cluster structure demonstrated that the 52 samples were separable into seven statistically different groups on the basis of community abundance composition of the 22 taxa (Figure 6b). (ii) CAP analysis identified the SAM as a significant explanatory variable on the structure of the phytoplankton community and





showed that identified clusters were generally distinguished by the SAM and *DaysAfter1Oct* (Table 1b, Fig. 6). (iii) 15 of the
22 taxa showed significant (p<0.05) pairwise correlations between relative abundance and at least one of the three SAM indices
(Table 2b). The greatest single influence in phytoplankton community composition was seasonal succession, as represented by
*DaysAfter1Oct*, which explained 15.4 % of variance in the multiparameter CAP model (Table 1b), however two modes of
SAM explained a further 16.9 % in total and will be discussed first.

The SAM mode with greatest influence on phytoplankton community composition, *SAM autumn* (Fig. 3, 4) explained 12.6
% of variance in the multiparameter CAP model. It represented the average SAM around the time that sea ice was extending
northward through the SIZ (Fig. 1a). At this time, phytoplankton productivity in the SIZ would have declined to around 30 %
of its mid-summer maximum (Moore and Abbott, 2000; Arrigo et al., 2008; Constable et al., 2014), and phytoplankton would
be preparing for winter by variously producing energy storage products, producing resting spores or cysts, reducing metabolic
rate, and engaging in heterotrophic consumption for energy (Fryxell, 1989; McMinn and Martin, 2013). The formation of sea
ice reduces available light by as much as 99.9 % (McMinn et al., 1999), severely limiting light for phytoplankton for more
than half a year: at the range of longitude sampled, latitude 64˚S was sea-ice covered for half the time across the sampled
years (Fig. 1a). Windier conditions associated with higher *SAM autumn* may delay the consolidation of sea ice into larger
floes (Roach et al., 2018), extending the phytoplankton growing season, and possibly increasing the relative abundance of taxa
that occur later in the season. This was supported by the observation that the only two taxa observed to have significantly
higher relative abundance later in the spring-summer, the *Chaetoceros* species *C. dichaeta* and *C. concavicornis/curvatus*,
were both observed to also show significantly higher relative abundances when the SAM in the preceding autumn was higher
(Table 2b). Higher SAM in the autumn is expected to result in deeper autumn mixed layers, reducing the photosynthetic
rate of individual phytoplankton cells as they cycle below the critical depth (Sverdrup, 1953; Sathyendranath and Browman,
2015), whilst conversely enhancing potential phytoplankton productivity by maintaining the input of nutrient rich deep water
to the euphotic zone. The quantity of phytoplankton that survive the Antarctic winter is extremely low (McMinn and Martin,
2013), and the abundance of taxa present when the sea ice forms may strongly influence the availability of phytoplankton
to seed the subsequent spring-summer bloom. Extending the productive season by delaying the consolidation of sea ice may
result in greater declines in relative abundance for taxa that are more prolific earlier in the season. Of the eight taxa showing
statistically (p<0.05) higher relative abundance earlier in the spring-summer, three showed corresponding statistically lower
relative abundances with higher preceding *SAM autumn* (*Emiliana huxleyi*, *Nitzschia acicularis/decipiens*, and *Parmales* spp.,
Table 2b), although four of the remaining five taxa showing no detectable relationship with *SAM autumn*.

Two other SAM modes were found to influence phytoplankton: *SAM spring* and *SAM prior*. These modes were difficult to
distinguish due to their largely overlapping time periods (Fig. 4), and they were strongly correlated (r: 0.83, Table 2c), with
similar influence on taxonomic abundances. *SAM prior* was the preferred parameter for the multiparameter CAP model, in
which it explained 4.3 % of total variance. Windier and stormier conditions associated with higher SAM in the months prior
to sampling would increase nutrient input to the euphotic zone from deeper waters (Lovenduski and Gruber, 2005), promot-
ing productivity, whilst at the same time episodically diluting surface phytoplankton through deeper mixing. More stormy





conditions may also have brought about a faster break-up of sea ice, promoting phytoplankton growth. Conversely, it would also restrict stratification of the surface ocean, precluding bloom formation, lessening productivity (Fitch and Moore, 2007)

and reducing the abundance of early blooming taxa. This may explain the responses of *Emiliania huxleyi* and the combined *Nitzschia acicularis/decipiens* group which both showed early maximum abundances and also negative correlations with *SAM spring* and *SAM prior* (Table 2b). Six other taxa with early maximum abundance (negative correlation with *DaysAfter1Oct*) showed no detectable correlation with *SAM spring*, indicating that their abundance was determined by environmental factors that prevail early in season but not those factors altered by variations in the SAM.

Historically, the variance in the SAM in the spring quarter is lower than in other quarters (NOAA 2005), perhaps explaining why *SAM spring* and *SAM prior* explained less variation in community composition than *SAM autumn*. The small *Chaetoceros neglecta* and the lightly silicified *Dictyocha speculum* both showed positive relationships with *SAM prior* but not with *SAM autumn* or *SAM spring*. Yet both these taxa also showed a strong influence of *DaysAfter1Oct* on their relative abundance and the strength of this relationship may have obscured any pairwise correlation with the SAM and other variables.

The SAM typically describes around 35 % of total observed Southern Hemisphere climate variability (Marshall, 2007). Hence only a third of any covariance between climate and phytoplankton community composition might be expressed as covariance between the SAM and community composition, and thus the variance in community composition due to variation in climate could well be greater than we detected with the SAM.

## 5.2 Taxa influenced by SAM

Nothing has been previously reported with respect to the climatic preferences of the majority of taxa identified in this study. Some of the observed taxa have been reported showing various relationships with environmental factors, including *SST*, time through the season, and latitude, but often at a genera rather than a species level (Burckle et al., 1987; Chiba et al., 2000; Waters et al., 2000; Green and Sambrotto, 2006; Gomi et al., 2007). We, however, observed different responses to environmental variables among closely related taxa. This was exemplified by the opposite correlations of *Chaetoceros* species *C. dicheata*

and *C. neglectus* with *DaysAfter1Oct* (0.48 and -0.70 respectively, Table 2b) and the opposite correlations of *Fragilariopsis* species *F. rhombica* and *F. kerguelensis* with *SAM autumn* (-0.39 and 0.52 respectively, Fig. 5a,b). The strong and opposite response to these variables by species belonging to the same genus indicates the importance of species-level observation in detecting subtle changes in pelagic phytoplankton communities.

The abundance of *Emiliania huxleyi*, the dominant coccolithophorid in the world's oceans (Cubillos et al., 2007), showed

a moderate negative relationship with all three identified SAM indices, and a weak negative relationship with *DaysAfter1Oct* (Table 2b). It also showed a moderate negative relationship with the year of sample collection, suggesting abundance declined in the SIZ over the study period. Cardinal et al. (2007) reported a near absence of coccolithophorids south of the Polar Front (latitude 55°S), and (Cubillos et al., 2007) reported surface-water *E. huxleyi* abundances declining southward through the SIZ





to near absence by 65°S. No variation in the relative abundance of this species with latitude was seen across the 62°S to 64.5°S
latitudinal range sampled.

## 5.3   The effects of SAM on biomass

Our study clearly showed that variation in the SAM coincided with variation in the structure of the phytoplankton community,
but we did not detect any influence on total estimated phytoplankton cell volume or volume-inferred phytoplankton biomass.
Positive SAM has previously been shown to be associated with increased standing stocks and productivity of phytoplankton in
the SIZ of the SO (Arrigo et al., 2008; Boyce et al., 2010; Soppa et al., 2016). In the SIZ above the Antarctic Divergence, nutri-
ents consumed by phytoplankton from surface waters through the spring and summer are replenished by deep-water upwelling
through the following winter. Thus, the levels of nutrition remaining at the end of summer integrate the total draw-down of
nutrients by phytoplankton production over the entire spring-summer growing season (Arrigo et al., 1999). We observed this
drawdown as the negative correlation between all nutrient concentrations and ***DaysAfter1Oct*** (Table 2d). We also observed
a negative relationship between ***SAM autumn*** and all macro-nutrient concentrations the following spring-summer (Table 2d)
suggesting that elevated SAM in autumn leads to greater productivity and thus greater nutrient drawdown during the following
spring-summer.

## 5.4   Sensitivity of phytoplankton taxonomic composition to climate change

We detected the effect of variation in the SAM on the composition of phytoplankton communities in the SIZ (Table 1, 2b), and
on their productivity as inferred from nutrient draw-down (Table 2d). Henson et al. (2010) estimated climate change driven
trends in chlorophyll and primary production would not become apparent in the SO until around 2055, as natural fluctuations in
these variables are large relative to the effect of global warming. The climate of the SO is more variable than climates of lower
latitudes due to interactions between atmosphere, ocean, and ice, making the detection of any signal of climate change difficult
(Turner et al., 2015). Although change in surface air temperature is already apparent at equatorial latitudes, changed surface
air temperature in the SIZ of the SO is not expected to be detectable until 2050 or later (Hawkins and Sutton, 2012). Whilst
our study did not show significant ($\alpha$=0.05) increase in ***SAM autumn*** or ***SAM spring*** over the 11 years sampled, a statistically
significant upwards trend of 0.01 SAM points per year has been seen over the period from 1979 through to at least 2014
(Arblaster and Meehl, 2006; Gillett and Fyfe, 2013; Jones et al., 2016). Nevertheless, the differing responses of phytoplankton
taxa to their environment, and the integrating effect of successional change, enabled change in phytoplankton composition
to be detected, suggesting that the phytoplankton composition is a more sensitive indicator of environmental change than the
direct temperature record.

## 5.5   Seasonal succession in taxonomic composition

Phytoplankton taxonomic composition was expected to follow a successional progression through the spring-summer (Lancelot
et al., 1993; Arrigo et al., 1999; Thompson et al., 2000; Davidson et al., 2010; Rigual-Hernández et al., 2015). Ten of the 22





taxa showed a significant relationship between relative abundance and ***DaysAfter1Oct*** (Table 2b), the foremost explanatory environmental variable, explaining 15.4 % of total observed variance in phytoplankton taxonomic composition (Table 1a). This variable likely represented a proxy for many important unmeasured processes such as solar radiation, increasing mixed-layer depth, and variations in grazing mortality. ***DaysAfter1Oct*** also covaries with measured variables that also exhibit seasonal changes, including ***Salinity***, ***DaysSinceSeaIce***, sea surface temperature (***SST***) and the concentrations of macronutrients. Thus,

it is unsurprising ***DaysAfter1Oct*** was the largest explanator of phytoplankton change. This variable, along with ***DaysSinceSeaIce*** and ***SST***, and the range of environmental variables that covary with these environmental factors, drive the seasonal succession of the phytoplankton, from near-surface blooms of large diatoms at the marginal ice edge as it recedes southward across the SIZ, to small diatoms and flagellates forming deep chlorophyll maxima once the nutrients have been depleted (Wright et al., 2010). ***DaysAfter1Oct*** has the limitation of being used as a linear variable in this analysis, potentially not de-

tecting influences on taxa that peak mid-season when both solar radiation and productivity are at a maximum (Arrigo et al., 1998).

The significant correlation observed between ***Long.E*** that a sample was collected and ***DaysAfter1Oct*** that it was collected indicates that the resupply voyages earlier in the season were further east when they traversed the SIZ. This could have been due to avoidance of pack ice or some other navigational consideration of the resupply voyages, however it confounds the

two variables and some of the variance attributable to ***DaysAfter1Oct*** may be due to geographic variation on the longitude of sampling. Longitude cannot be considered an absolute variable in temporal studies of surface water in the SIZ of the SO, as surface water is moving. Surface water north of the Antarctic Divergence (AD) has been recorded moving west to east at velocities in the order of 15 cm s$^{-1}$, and south of the AD, east to west at similar velocities (Bindoff et al., 2000). At this velocity, surface water would completely cross the 625 km of longitude sampled in the current study in 48 days (samples were collected

over 131 calender days), and surface water sampled late in the spring-summer may have been 1000 km to the east or west early in the spring-summer period. Further, surface water north of the AD has a northward component to its movement and surface water south of the AD has a southward component to its movement (Bindoff et al., 2000; Wilkins et al., 2013), and thus latitude is also confounded with ***DaysAfter1Oct***, although velocities are much lower and the correlations observed here were not statistically significant (Extra Material S1).

The time since sea ice retreat has previously been identified as an important covariate for explaining phytoplankton population dynamics in the SIZ (Thompson et al., 2000; Garibotti et al., 2005; Davidson et al., 2010; Wright et al., 2010). In this study, the ***DaysSinceSeaIce*** showed pair-wise relationships with abundances of taxa that were similar to those we observed for ***DaysAfter1Oct***, although singly explaining less of the variance in taxonomic composition (5.9 % versus 15.4 % respectively - Table 1a). This difference was also observed in the relative abundances of individual taxa. For example, the relative

abundance of *Chaetoceros dichaeta* showed a positive relationship with ***DaysAfter1Oct*** (r: 0.48, Table 2b) and a lesser relationship with ***DaysSinceSeaIce*** (r: 0.37), while the relative abundance *Chaetoceros neglectus* showed a negative relationship with ***DaysAfter1Oct*** (r: -0.70) and lesser relationship with ***DaysSinceSeaIce*** (r: -0.40).





### 5.6 Taxa not influenced by the SAM

A third of analysed taxa, comprising 7 taxa and 23 % of all counted cells, showed no detectable relationship with the SAM.
This could be due to large errors associated with low counts of rarer taxa, because unaccounted variation was masking any
relationship, or because the taxa were insensitive to the SAM. There is less chance of detecting relationships between taxa and
environment variables when fewer individuals are counted, however some less represented taxa (e.g. *Emiliania huxleyi*) did
show relationships with SAM indices.

Five of the 22 taxa resolved showed no significant relationships with either the SAM or ***DaysAfter1Oct***. All were compara-
tively scarce and together represented only 2 % of all cells counted. Assessing species compositions across a greater fraction
of each sample, and thus counting more of the scarcer taxa, may have revealed relationships between these rarer taxa and envi-
ronmental variables (Nakagawa and Cuthill, 2007). Yet it remains possible that these taxa are actually unaffected by seasonal
succession and the SAM, instead responding to other environmental variables that were not measured as part of this study, or
that they remain as a persistent but relatively rare background taxa with respect to the overall phytoplankton assemblage.

### 5.7 The first study relating the SAM to phytoplankton taxonomic composition

This is the first study to show an effect of changes in the SAM index on the composition of phytoplankton communities in the
SO, although such findings have already been reported for other major climatic phenomena. The climatically similar Northern
Hemisphere Annular Mode (NAM) causes increased westerly winds and deeper mixed layers at mid- to high northern latitudes
in its positive phase (Nehring, 1998; Thompson et al., 2003; Kahru et al., 2011). The NAM has been related to the timing,
abundance and biomass of phytoplankton taxa at high northern latitudes (Nehring, 1998; Belgrano et al., 1999; Ottersen et al.,
2001; Blenckner and Hillebrand, 2002), and to delayed time of maximum chlorophyll in the North Atlantic Summer (Kahru et
al., 2011). Similarly, the El Niño Southern Oscillation (ENSO) equatorial mode has been shown to influence the distribution
and abundance of phytoplankton in the tropical oceans (Blanchot et al., 1992).

### 5.8 Implications

The SIZ is a productive region of the SO (Moore and Abbott, 2000), and changes to the SIZ phytoplankton community have
potentially far-reaching implications for the ecosystem services these organisms provide, including carbon sequestration and
supporting the productivity of almost all Antarctic life. Increases in the relative abundance of the larger *Chaetoceros* spp.
diatoms would favour grazing by large metazooplankton, especially krill (Boyd et al., 1984; Kawaguchi et al., 1999; Moline et
al., 2004), which link phytoplankton to whales, seabirds, seals, and most higher Antarctic life forms (Smetacek, 2008). Such
changes would also increase the efficiency of the biological pump as the larger phytoplankton sink more rapidly than small
(Alldredge and Gotschalk, 1989), and increased grazing by krill would reparcel the phytoplankton cells into faeces that would
also sink fast (Cadée et al., 1992). Such changes in carbon flux and trophodynamics would act as a negative feedback on climate
change by speeding the sequestration of carbon in the deep ocean.

Phytoplankton are the pastures of the oceans and it is not surprising that the climate in both autumn and spring influence
the taxonomic composition of phytoplankton and their ecological progression through the productive spring-summer period in
the SIZ. Climate change impacts have now been documented across every type of ecosystem on Earth (Scheffers et al., 2016;
Harris et al., 2018) and the distribution, abundance, phenology and productivity of phytoplankton communities throughout
the world are changing in response to warming, acidifying, and stratifying oceans (Hoegh-Guldberg and Bruno, 2010). The
surprise is that changes in the taxonomic composition associated with the SAM were detectable over a relatively brief eleven-
year monitoring period and despite all the other environmental factors that elicit variability in phytoplankton communities.

The SAM is predicted to become increasingly positive in the future (Arblaster and Meehl, 2006; Swart and Fyfe, 2012;
Gillett and Fyfe, 2013; Abram et al., 2014; Solomon et al., 2016). Our results cannot necessarily be extrapolated to infer
changes that will likely occur as the SAM continues to increase, as evolutionary responses can partly mitigate adverse effects
on phytoplankton of longer-term climate change, and future climate changes are likely to impose other co-stressors on phyto-
plankton inhabiting these waters (Lohbeck et al., 2014; Schlüter et al., 2014; Deppeler and Davidson, 2017). The present study
demonstrates, for the first time, that variation in the SAM influences the taxonomic composition of phytoplankton in the SIZ
of the SO. The relationships between the SAM and community composition were complex but significant, and the degree of
observed covariance warrants further investigation.

**6 Conclusions**

We found that the Southern Annular Mode was influential on phytoplankton community composition in the seasonal ice zone
of the Southern Ocean, second only to the seasonal succession variable (***DaysAfter1Oct***). This influence suggests that the
phytoplankton of the SIZ are indeed susceptible to changes in the SAM and thus possibly to climate change.

*Data availability.* https://data.aad.gov.au/metadata/records/SAM_influences_phytoplankton_community_composition

*Author contributions.* Bruce L. Greaves: Conceptualization, Data curation, Formal analysis, Investigation, Methodology, Software, Super-
vision, Validation, Visualization, Writing – original draft, Writing – review & editing Andrew T. Davidson: Conceptualization, Funding
acquisition, Formal analysis, Methodology, Project administration, Resources, Supervision, Writing – review & editing Alexander D. Fraser:
Formal analysis, Methodology, Resources, Writing – review & editing John P. McKinlay: Formal analysis, Methodology, Software, Writing
– review & editing Andrew Martin: Project administration, Supervision, Writing – review & editing Andrew McMinn: Funding acquisition,
Project administration, Resources, Writing – review & editing Simon W. Wright: Conceptualization, Funding acquisition, Formal analysis,
Writing – review & editing



*Competing interests.* The authors declare that they have no conflict of interest.

*Acknowledgements.* Sampling on Astrolabe was supported by a French-Australian research collaboration. The Institut Polaire Français Paul-Émile-Victor supported access to the ship and field operations. The underway biogeochemical data collection was coordinated by Prof Alain
Poisson and Dr Nicolas Metzl, Sorbonne Université, and Dr Bronte Tilbrook, CSIRO Oceans and Atmosphere. Steve Rintoul (CSIRO) and Rose Morrow (LEGOS) coordinated the collection of underway Salinity and temperature data. The Antarctic Climate and Ecosystems CRC and the Integrated Marine Observing System are thanked for supporting the operation of underway sensors, the collection of water samples and analysis of nutrient analyses reported in this study. Alan Poole, Matt Sherlock, John Akl, Kate Berry, Lesley Clementson, Brian Griffiths (CSIRO), Rick van den Enden and Rob Johnson (AAD) and the many dedicated volunteers and ships' officers and crew are thanked for
their important contributions to the field efforts and data management. We thank the University of Tasmania and the Australian Antarctic Division for the space and resources needed to undertake this work. Thanks to Prof. Nathaniel Bindoff and Dr Simon Wotherspoon for their consideration of parts of the manuscript. This work was supported by the Australian Government's Cooperative Research Centre program through the Antarctic Climate & Ecosystems CRC, the Australian Antarctic Division (Project 40 and 4107), and by the Australian Research Council's Special Research Initiative for Antarctic Gateway Partnership (Project ID SR140300001).



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





**Table 2.** (a) Summary statistics for environmental variables; (b) correlations between taxa relative abundances and environmental covariates; (c) correlations between environmental covariates; (d) correlations between macro-nutrient concentrations and environmental covariates. Correlations significant at p≤0.05 are in bold italics, correlations significant after Bonferroni adjustment are also underlined (p<0.05/19 for correlations between environmental variables, p<0.05/20 for correlations with taxa relative abundance).

| | cells counted | DaysAfter1Oct | SAM autumn | SAM prior | SAM spring | Long.E | DaysSinceSeaIce | SST | Salinity | year |
|---|---|---|---|---|---|---|---|---|---|---|
| **(a) Statistics for environmental covariables** | | | | | | | | | | |
| unit | | days | index | index | index | °E | days | °C | PSU | year |
| average | | 96 | -0.2 | 0.1 | 0.4 | 142 | 65 | 0.6 | 33.7 | - |
| min | | 20 | -0.8 | -1.3 | -1.5 | 136 | -26 | -1.8 | 33.2 | 2002 |
| max | | 151 | 0.6 | 2.0 | 10.0 | 148 | >365 | 3.0 | 34.1 | 2012 |
| n | | 52 | 11 | 52 | 11 | 52 | 52 | 5 | 52 | 11 |
| average standard error of estimate | | - | 0.14 | 0.13 | 0.14 | - | - | - | - | - |
| **(b) Correlations with taxa relative abundance** | | | | | | | | | | |
| *Chaetoceros atlanticus* | 356 | -0.15 | ***0.55*** | ***0.57*** | ***0.63*** | 0.20 | -0.01 | -0.20 | 0.22 | 0.13 |
| *Chaetoceros castracanei* | 48 | ***-0.36*** | -0.02 | 0.26 | 0.20 | ***0.41*** | -0.12 | ***-0.36*** | -0.07 | -0.07 |
| *Chaetoceros concavicornis/curvatus* | 120 | ***0.37*** | ***0.36*** | 0.27 | ***0.35*** | -0.07 | 0.27 | 0.25 | -0.14 | 0.11 |
| *Chaetoceros dichaeta* | 2563 | ***0.48*** | ***0.38*** | ***0.31*** | ***0.29*** | -0.13 | ***0.37*** | ***0.35*** | -0.17 | 0.20 |
| *Chaetoceros neglectus* | 634 | ***-0.70*** | -0.06 | ***0.42*** | 0.24 | ***0.48*** | -0.40 | ***-0.69*** | ***0.56*** | -0.04 |
| *Cylindrotheca closterium* | 122 | 0.13 | 0.09 | -0.10 | -0.03 | 0.02 | 0.32 | 0.12 | 0.02 | -0.11 |
| *Dactyliosolen antarcticus* | 277 | 0.18 | ***0.37*** | ***0.34*** | ***0.27*** | -0.06 | 0.18 | 0.13 | -0.08 | 0.06 |
| *Dactyliosolen tenuijunctus* | 1981 | -0.18 | ***-0.44*** | -0.08 | -0.16 | 0.16 | -0.19 | -0.17 | 0.23 | -0.02 |
| *Dictyocha speculum* (silicoflagellate) | 81 | ***-0.78*** | -0.17 | ***0.30*** | 0.14 | ***0.68*** | ***-0.41*** | ***-0.75*** | ***0.36*** | -0.14 |
| discoid centric diatoms | 959 | ***-0.57*** | 0.15 | 0.06 | 0.24 | ***0.52*** | -0.11 | ***-0.57*** | 0.21 | -0.15 |
| *Fragilariopsis cylindrus/curta* | 3987 | 0.26 | -0.06 | -0.08 | -0.09 | ***-0.58*** | -0.08 | ***0.35*** | -0.12 | 0.24 |
| *Fragilariopsis kerguelensis* | 1031 | 0.23 | ***0.52*** | 0.16 | 0.25 | -0.07 | 0.19 | 0.22 | ***-0.46*** | -0.05 |
| *Fragilariopsis pseudonana* | 170 | -0.13 | 0.22 | -0.02 | 0.22 | -0.10 | -0.05 | -0.03 | 0.12 | 0.22 |
| *Fragilariopsis rhombica* | 4542 | 0.16 | ***-0.39*** | ***-0.58*** | ***-0.57*** | -0.13 | 0.13 | 0.22 | -0.12 | -0.24 |
| *Fragilariopsis ritscheri* | 46 | 0.11 | -0.10 | 0.00 | -0.03 | -0.02 | 0.02 | 0.10 | -0.03 | 0.03 |
| *Guinardia cylindrus* | 110 | 0.09 | 0.12 | -0.06 | -0.06 | 0.05 | 0.17 | 0.10 | -0.03 | -0.02 |
| *Emiliania huxleyi* (haptophyte) | 173 | ***-0.28*** | ***-0.38*** | ***-0.42*** | ***-0.38*** | 0.21 | 0.12 | -0.25 | -0.01 | ***-0.37*** |
| *Nitzschia acicularis/decipiens* | 1133 | ***-0.47*** | ***-0.45*** | -0.29 | ***-0.31*** | ***0.42*** | ***-0.32*** | ***-0.46*** | 0.09 | -0.22 |
| *Parmales* spp. (chrysophyte) | 322 | ***-0.60*** | -0.29 | 0.15 | -0.09 | ***0.42*** | ***-0.42*** | ***-0.65*** | ***0.36*** | ***-0.28*** |
| *Petasaria heterolepis* | 45 | -0.25 | -0.13 | ***-0.27*** | -0.08 | 0.15 | -0.17 | -0.25 | 0.02 | -0.02 |
| *Pseudo-nitzschia lineola* | 681 | ***-0.35*** | ***0.39*** | 0.19 | ***0.37*** | ***0.36*** | -0.09 | ***-0.35*** | 0.18 | 0.01 |
| *Thalassiothrix antarctica* | 112 | -0.16 | ***0.32*** | 0.12 | 0.16 | 0.15 | -0.11 | -0.11 | -0.19 | -0.15 |





|  | Environmental covariables | | | | | | | | |
|---|---|---|---|---|---|---|---|---|---|
|  | *DaysAfter1Oct* | *SAM autumn* | *SAM prior* | *SAM spring* | *Long.E* | *DaysSinceSeaIce* | *SST* | *Salinity* | *year* |
| (c) Correlations amoung environmental variables | | | | | | | | | |
| *SAM autumn* | *0.32* | | | | | | | | |
| *SAM prior* | -0.06 | _**0.51**_ | | | | | | | |
| *SAM spring* | 0.04 | 0.56 | _**0.83**_ | | | | | | |
| *Long.E* | _**-0.63**_ | -0.17 | 0.10 | 0.05 | | | | | |
| *DaysSinceSeaIce* | _**0.56**_ | 0.18 | -0.03 | 0.07 | -0.27 | | | | |
| *SST* | _**0.92**_ | 0.27 | -0.14 | -0.03 | _**-0.68**_ | _**0.60**_ | | | |
| *Salinity* | _**-0.43**_ | -0.14 | *0.31* | 0.21 | 0.23 | -0.13 | _**-0.41**_ | | |
| *year* | 0.18 | 0.27 | *0.35* | 0.32 | -0.24 | 0.02 | 0.27 | -0.06 | |
| (d) correlations with macro-nutrients (n=51) | | | | | | | | | |
| [NOx] | _**-0.77**_ | _**-0.39**_ | 0.23 | 0.04 | _**0.53**_ | _**-0.43**_ | _**-0.72**_ | _**0.54**_ | -0.14 |
| [PO4] | _**-0.73**_ | _**-0.56**_ | -0.07 | -0.26 | _**0.62**_ | _**-0.52**_ | _**-0.70**_ | *0.39* | -0.13 |
| [SiO4] | _**-0.56**_ | _**-0.42**_ | 0.26 | -0.05 | *0.40* | _**-0.49**_ | _**-0.63**_ | *0.39* | 0.09 |