# Peer review of "The Southern Annular Mode (SAM) influences phytoplankton communities in the seasonal ice zone of the Southern Ocean"

_Biogeosciences, 2019_

## Referee Comment (RC1) · Anonymous Referee #1 · 16 Nov 2019

In this manuscript entitled "The Southern Annular Mode (SAM) influences phytoplankton communities in the seasonal ice zone of the Southern Ocean", the authors examine the role of SAM on phytoplankton communities in the SIZ of the Southern Ocean. I think the document is not yet ready to be published, although the subject and results are really interesting. The structure of the document is really difficult to follow at the moment. I have listed some improvements that could be made to improve the clarity of the manuscript.

General comments: My main concern is related to the structure of the document, to many subsections, particularly in the sections on results and discussion (8 subsections

for discussion, and 2 sentences for conclusion, 1 sentence in the section on results (3.1). The document, as it is now, is unbalanced and difficult to read and needs to be reorganized around major themes (seasonal, interannual variability and impact on phytoplankton communities for example for the discussion).

In this paper, the authors examined the role of SAM and seasonal variability on changes in phytoplankton communities, but some key environmental factors are really missing in this study, (1) mixing estimates (by estimating the depth of mixed layers, deriving wind stress) and (2) light measurements (in situ or satellite data)? Because it can be suspected that changes in the intensity of the SAM will directly influence light-mixing regimes, and therefore changes in the composition of phytoplankton communities at the time of sampling? This is particularly important given that the authors mention the interaction between mixing and phytoplankton dynamics in the discussion.

In addition, the authors focused on understanding changes in the relative abundance of the main phytoplankton groups, but we have no idea how phytoplankton biomass could change annually with the SAM. The authors mentioned this briefly in the discussion (5.3), but can you access to any vertically integrated biomass proxies (vertically integrated chlorophyll, PP and satellite-derived estimates)?

This comment is related to the last one, but we have no idea where we stand with respect to phytoplankton phenology. In Figure 1, it would be nice to have satellite-derived time series of chlorophyll a, for example. The problem I see here is that the SAM could perhaps also change the phytoplankton phenology (bloom duration or timing for example). And perhaps what the authors have defined as interannual variability driven by the SAM can simply be related to a sampling of different phenological states. It would be important for me to check this point.

Specific comments: l.186-186: Can you add a table in the paper or in the supplementary materials listing these taxa (the 4 in all the samples, and the 11 in 90% of the samples)?

Table: Table 1: Long.E is indicated two times as variable, is it an error?

---

## Referee Comment (RC2) · Damiano Righetti (Referee) · 20 Nov 2019

Review for: Biogeosciences Discussions

Title: The Southern Annular Mode (SAM) influences phytoplankton communities in the seasonal ice zone of the Southern Ocean
Authors: Greaves et al.
1st review

Summary

Greaves et al. hypothesize that a large-scale climate mode (the Southern Annular Mode; SAM) shapes phytoplankton communities in the sea ice zone of Antarctica through effects on ocean mixing, nutrient upwelling or sea ice cover. This sea ice zone is relevant to marine primary productivity and marine carbon export and constitutes an exceptionally dynamic and cold habitat. To quantify SAM, the authors use an index, defined as the zonal mean sea level air-pressure at 40°S minus the zonal mean sea level air-pressure at 65°S (Gong and Wang, 1999). This index has become more positive throughout the years 1979 to 2017 (i.e., weaker high-latitude pressure, relative to the mid-latitude pressure) and was linked to stronger winds and upwelling at 60°-70°S (latitudes of the sea ice zone). The authors use a 'time-window approach' to detect the possible imprint of SAM on phytoplankton community composition, and they study composition-environment relationships for an additional 13 variables (including sea surface temperature, nitrate concentration, and sampling day, among others). The authors report that a time-averaged SAM signal explains 13.3% of variance in phytoplankton community composition across 52 samples (spanning 22 taxa and 11 spring/summer periods from 2002/3 through 2012/3). This SAM signal is obtained by averaging daily SAM across two months and by centering it at March 11[th] prior to the sampling period. However, the most powerful predictor reported is sampling day (15.4% of variance explained). Furthermore, the authors report disparate responses in the relative abundance of small vs. large diatom species to increasing (time-averaged) SAM. It is concluded that SAM signals influence phytoplankton communities in the seasonal ice zone of Antarctica.

General evaluation

This study focuses on the intersection of sea ice and water in Antarctica. From an ecological and climate point of view, the quantification of patterns and predictors of phytoplankton composition in this large-scale habitat appears timely and important. The data collected appear useful to address this understudied habitat. However, I identified a serious lack of clarity in the writing and structure in many parts of the manuscript (i), and have a main conceptual critique point (ii).

(i) Key concepts (SAM or SAM index) are not clearly defined. The SAM definition leaves it open to the reader, how the sign of the SAM index is calculated, and whether atmospheric pressure or water pressure constitutes the SAM index. There is a problem with clarity of statements and consistency of word use (e.g., different expressions are used for the same thing), and a lack of clear correspondence between hypothesis, methods, and key results. I provide detailed examples on clarity below.

(ii) My main conceptual critique point is that the impact of the time-averaged SAM signal in autumn on phytoplankton community composition in spring to summer has

not been firmly tested by the data shown. The study demonstrates that it is possible to average the daily SAM index in a way that a significant part of the variation in community composition can be explained in next spring/summer, yet it is unclear why microbial species that live on timescales from days to weeks, would respond to the SAM signal with a time-lag of several months. I suggest that relationships between a more positive state of SAM in autumn and temperature, wind speed, mixed-layer depths, and nutrient levels in spring to summer—factors that may directly shape phytoplankton composition—shall be evaluated, to support the paper's message. In section 4 ('Other relationships'), there are several relationships presented between predictors, yet the results are not presented in a structured way to support the hypothesis that SAM-induced changes in temperature, wind-speed, mixed-layer depth or nutrient concentrations affect community composition. The current association between the SAM signal (or "SAM modes") described and community composition may not be causal. In the context of fast-lived organisms it seems crucial to test if the link between summer community composition and (preceding) SAM is plausible.

Recommendations

I suggest that the manuscript is thoroughly screened for clarity. Second, besides further testing the associations of the SAM signal of autumn with physicochemical factors known to affect phytoplankton composition (and whether these associations are in line with expectation), I suggest splitting the 22 taxa into ecological test groups, which are expected to respond differently to changing mixing-, wind-, and nutrient-patterns under a more positive SAM state. These expectations can be presented as specific hypotheses in the introduction. Such a biological approach has been partly implemented by comparing small diatoms (presumably better adapted to stable waters) with large diatoms (presumably better adapted to strong mixing). Yet the results of this test lack a graphical presentation in the manuscript, across all taxa. Species may be grouped further into warm, temperate, or polar species, depending on their global distributions (e.g. using observations from OBIS and GIBF; Righetti et al., 2019) and their responses may differ under SAM-induced warming/cooling. Similarly, R-strategist (fast growing, light stress tolerant species) and S-strategists (slow growing, nutrient stress tolerant species) may be grouped together (Brun et al., 2015), as they may respond oppositely to changing nutrient levels. Additionally, species with large vs. small cells may show opposite responses to changing turbulence and wind regimes (Margalef, 1997, 1978). Finally, predicting the response of siliceous vs. calcareous taxa to SAM constitutes an exciting hypothesis: these groups have shown opposite responses to deeper mixing or nutrient entrainment (Cermeño et al., 2008). With respect to the clustering techniques used to describe communities I cannot give detailed recommendations, as the metrics used are beyond my expertise.

Detailed comments

There are too many comments to be listed. I therefore give examples for selected paragraphs, with comments on clarity, for each:

    Abstract:
- Line 3: How many variables were tested?
- Line 6: How many species (genera, higher taxa) were included among the 22 taxa?
- Line 7: I do not understand 'CAP'. This term has not been introduced.

- Lines 8, 9, 11, 17: The following terms are used: taxonomic community composition, taxa composition, phytoplankton community structure, taxonomic composition of phytoplankton. While I understand that the authors strive to include stylistic variation, the reader is confused by the multiple expressions. Do they denote the same thing or not? I recommend using use the same expression for the same thing. Else, once an expression is clear, an abbreviation of the latter may be used therein, as long as it denotes the same thing.
- Line 10: Unclear to me, if the correlation is significant or not.
- Line 13: Unclear to me, if "response" means a response of abundance or not.
- Line 15: Before, the expression "SAM index" was used, not "higher SAM". Does "higher SAM" refer to a more positive state of the SAM index?
- Line 17: Confusing, as taxonomic composition of phytoplankton is not the same thing as a standing stock (or a "pasture") of biomass of phytoplankton.
- Line 16: It is unclear to me, if the expression "pelagic ecosystem" is suitable in the context of a sea ice transition zone.
- Line 16: It is unclear, how many of the total species that were studied, responded significantly to SAM. Thus, it is unclear, if this result is important or general.
- Line 10 ff: It is surprising that 'day of sampling' explains more variation in community composition than any other locally sampled environmental factor (SST, nutrients, etc). An interpretation on why this is the case would help the reader to assess the plausibility or importance of this result.

Introduction:
- Line 21-23: The first two sentences are partially repetitive.
- Line 21 ff: The paragraph wants to establish the importance of phytoplankton productivity in the study area for global phytoplankton productivity. While the reader understands that a larger fraction (~30%) of carbon fixed by phytoplankton is exported in the study region, relative to the global average (~ 20% exported) it remains unclear, if the study region is globally important. What is the area-weighted contribution of the study region to global phytoplankton C-export?

1.2 The Southern Annular Mode
- Line 58 ff: Clarify the definition of SAM (see above). The reader cannot grasp how the sign of SAM is calculated or linked to changing pressure gradients, and thus how it is associated with physicochemical changes in the study system.
- Line 64 ff: SAM vs. SAM index vs. SAM state vs. SAM mode. Please use consistent expressions throughout the manuscript. (In addition, "taxon" could always refer to both a species and a group of species, and the use of "mode" in both the context of SAM and community composition may confuse the reader).

2.1 Phytoplankton composition and abundance
- Line 116: One reads as if the abundance of phytoplankton communities was sampled. As much as I understand, the abundance of species or taxa was sampled. (Then, an abundance-weighted community composition was calculated?).

2.3 Statistical analysis
- Line 151 ff: The methods section needs clarification, structurally and through editing. In this section, I have difficulties to understand whether three or more sets of analyses were performed based on the phytoplankton field data, and which of

these analyses is most important to test the key hypothesis of the paper, and at what temporal resolution the analyses were performed.
- Line 152: Has "community structure" really been correlated to "environmental covariates"? If I understand correctly, the abundance data was related to possible environmental drivers, per species. In this case, please specify: e.g. …and *species abundance between samples*…
- Line 151 ff: It is not clearly motivated, why clustering of community-level samples is suitable to identify the effect of SAM on community composition. To me, the number of 52 samples seems rather low already, and each degree of freedom may be valuable.

3. Results
- The first results presented to the reader are abundance-distributions of taxa across samples. Yet, the reader might expect that the most important piece of evidence to elucidate the role of SAM for phytoplankton composition is first presented.
- Line 206 ff. Can $P$, n, and $R^2$-values be provided for the correlations?
- Table 1: I do not understand, why nutrients are excluded in this table.
- Figure 5. The caption remains vague. What are the "several underlying assumptions" of linear regression? Relevant to be discussed in the caption?

Overall, the manuscript requires a clear structure in order to show to what degree the SAM signals may matter to community composition, based on (ecological) hypotheses tested and data. The support in the data for this message, and the evaluation of the manuscript are complicated at current and warrant further attention.

Damiano Righetti, November 20, 2019

References

Brun, P., Vogt, M., Payne, M.R., Gruber, N., O'Brien, C.J., Buitenhuis, E.T., Le Quéré, C., Leblanc, K., Luo, Y.-W., 2015. Ecological niches of open ocean phytoplankton taxa. Limnol. Oceanogr. 60, 1020–1038. https://doi.org/10.1002/lno.10074

Cermeño, P., Dutkiewicz, S., Harris, R.P., Follows, M., Schofield, O., Falkowski, P.G., 2008. The role of nutricline depth in regulating the ocean carbon cycle. Proc. Natl. Acad. Sci. U. S. A. 105, 20344–9. https://doi.org/10.1073/pnas.0811302106

Gong, D., Wang, S., 1999. Definition of Antarctic oscillation index. Geophys. Res. Lett. 26, 459–462. https://doi.org/10.1029/1999GL900003

Margalef, R., 1997. Margalef - 1997 - Turbulence and marine life.pdf.

Margalef, R., 1978. Life forms of phytoplankton as survival alternatives in an unstable environment. Oceanol. Acta 1, 493–509. https://doi.org/https://archimer.ifremer.fr/doc/00123/23403/

Righetti, D., Vogt, M., Zimmermann, N.E., Gruber, N., 2019. Phytobase: A global synthesis of open ocean phytoplankton occurrences, Earth Syst. Sci. Data Discuss. Earth Syst. Sci. Data Dicussions. https://doi.org/https://doi.org/10.5194/essd-2019-159

---

## Author Comment (AC1) · 23 Dec 2019

Response to comments on the submitted manuscript: Greaves et al. - *The Southern Annular Mode (SAM) influences phytoplankton communities in the seasonal ice zone of the Southern Ocean*

23 December 2019
* * *
We thank the reviewer for their valuable feedback on this manuscript. These have identified several areas for improvement of the manuscript, which we have addressed below:

**RC1 - Anonymous Referee #1, 16 November 2019**

**In this manuscript entitled "The Southern Annular Mode (SAM) influences phytoplankton communities in the seasonal ice zone of the Southern Ocean", the authors examine the role of SAM on phytoplankton communities in the SIZ of the Southern Ocean.**

**I think the document is not yet ready to be published, although the subject and results are really interesting.**

- Certainly, the results are very interesting

**The structure of the document is really difficult to follow at the moment.**

- We will carefully review and improve the structure of the manuscript in reference to the comments of both reviewers

**I have listed some improvements that could be made to improve the clarity of the manuscript.**

**General comments: My main concern is related to the structure of the document, to many subsections, particularly in the sections on results and discussion (8 subsections for discussion, and 2 sentences for conclusion, 1 sentence in the section on results (3.1). The document, as it is now, is unbalanced and difficult to read and needs to be reorganized around major themes (seasonal, interannual variability and impact on phytoplankton communities for example for the discussion).**

- We will refine manuscript structure

**In this paper, the authors examined the role of SAM and seasonal variability on changes in phytoplankton communities, but some key environmental factors are really missing in this study, (1) mixing estimates (by estimating the depth of mixed layers, deriving wind stress)**

- We don't have this information for each sample, or for the time periods prior – we are surmising that SAM influences wind-speed and subsequently mixed-layer-depth from the previously published observed and predicted positive relationship between the SAM and wind speed.

**and (2) light measurements (in situ or satellite data)?**

- We don't have this information for each sample

**Because it can be suspected that changes in the intensity of the SAM will directly influence light-mixing regimes, and therefore changes in the composition of phytoplankton communities at the time of sampling?**

**This is particularly important given that the authors mention the interaction between mixing and phytoplankton dynamics in the discussion.**

- It has been previously reported that SAM has been observed and predicted to relate to wind intensity – thus we used this to help explain how the identified maxima in SAM relationship with phytoplankton taxonomic composition could be plausible

**In addition, the authors focused on understanding changes in the relative abundance of the main phytoplankton groups, but we have no idea how phytoplankton biomass could change annually with the SAM.**

- Previous researchers have concluded that long term changes in the SAM will influence productivity: "*Lovenduski and Gruber (2005) predicted that increased SAM would support higher phytoplankton productivity, and subsequent analyses by Arrigo et 90 al. (2008); Boyce et al. (2010), and Soppa et al. (2016) have confirmed a positive relationship between the SAM and phytoplankton standing stocks and productivity south of 60°S in the SIZ*" (from line 88)
- We will add NASA satellite total chlorophyll estimates which we had been able to obtain for 49 of the 52 samples, which also show a positive relationship with SAM, i.e. higher SAM is associated with higher NASA satellite total chlorophyll (new Table 3 below, was Table 2 in previous manuscript)
- The peak of SAM influence in the preceding autumn was also detected in response surfaces for NASA satellite total chlorophyll (correlation between SAM in autumn and NASA total chlorophyll is 0.5) and nutrient levels (correlation between SAM in autumn and [$PO_4$] was -0.64 for all samples, and -0.84 for the later-in-season half of the samples) – these response surfaces will be included in the extra material (as drafted below). NASA satellite total chlorophyll and [$PO_4$] are observationally independent of the taxonomic counts, so similar prior-autumn maxima for the correlation with SAM and these traits are supportive of our finding that "*time-averaged SAM signal in autumn influences phytoplankton community composition in spring to summer*"

[Figure]

*Figure [Supplementary material]: Response surfaces of the correlation between NASA satellite total chlorophyll and the averaged SAM, versus timing and length of the SAM period. The SAM period is the number of days of daily-SAM averaged (vertical axis) and the timing of the range of averaged daily-SAM (horizontal axis). The SAM maxima identified in Figure 3 are shown (SAM autumn, SAM spring). Evident maxima in autumn are indicated with red broken line loops. (a) Analysis includes all available data (n=51), (b) analysis includes only half of the samples, being those collected later in the spring-summer productive season (n=26).*

**The authors mentioned this briefly in the discussion (5.3), but can you access to any vertically integrated biomass proxies (vertically integrated chlorophyll, PP and satellite-derived estimates)?**

- The last paragraph in the (existing) results section states that total volume and inferred biomass was estimated but not found to be related to SAM: "*Neither relative taxonomic total cell volume, estimated using the method of Hillebrand et al. (1999), or inferred relative 275 taxonomic total cell biomass, estimated using the method of Menden-Deuer and Lessard (2001), showed influence of any of the SAM indices (results not shown)*." (line 274)

- The only productivity effect was that inferred from nutrient drawdown, which showed reduced nutrients with more positive prior SAM indices, with the relationships with prior SAM indices (**SAM spring**, **SAM prior**, and **SAM autumn**) all being stronger when only the samples collected later in the season (the later half of samples) were included. In the SIZ of the Southern Ocean, surface-water nutrition is replenished through the winter by upwelling of deep ocean water at the Antarctic Divergence. The nutrient contents later in the spring-summer better reflect the total production over the spring-summer than do all samples, including those collected earlier in the spring-summer (as tabulated in the new Table 3 below). We will include the response surfaces for the correlation between the SAM and [$PO_4$] [depicted below] in Supplementary Material.

[Figure]

*Figure [Supplementary material]: Response surfaces of the correlation between [PO₄] and the averaged SAM, versus timing and length of the daily-SAM averaging range, i.e. the calendar date of the mid-point of the date range (horizontal axis), and the number of days over which those indices were averaged (vertical axis), respectively. The SAM maxima identified in Figure 3 are shown (SAM autumn, SAM spring). Evident maxima in autumn are indicated with red broken line loops. (a) Analysis includes all available data (n=51), (b) analysis includes only half of the samples, being those collected later in the spring-summer productive season (n=26).*

- More positive SAM in the prior autumn may lengthen the prior productive season, resulting in greater nutrient drawdown in the prior productive season, which might reduce the degree to which nutrients are replenishment through the winter? – we will consider this in the discussion.
- We will add NASA satellite total chlorophyll estimates which we had been able to obtain for 49 of the 52 samples, which also show a positive relationship with SAM, i.e. higher SAM is associated with higher NASA satellite total chlorophyll (new Table 3 below, was Table 2 in previous manuscript)

| | DaysAfter10Oct | SAM_autumn | SAM_prior | SAM_spring | Long.E | DaysSinceSeaIce | SST | Salinity | year | NASA.chla |
|---|---|---|---|---|---|---|---|---|---|---|
| unit | days | index | index | index | °E | days | °C | PSS | austral summer | WHAT |
| average | 96 | -0.04 | 0.06 | -0.16 | 142.0 | 65 | 0.63 | 33.7 | - | 0.29 |
| min | 20 | -0.66 | -1.35 | -1.49 | 135.8 | -26 | -1.80 | 33.2 | 2002 | 0.07 |
| max | 151 | 0.62 | 1.96 | 1.14 | 147.9 | 366 | 2.98 | 34.1 | 2012 | 0.70 |
| n | 52 | 11 | 52 | 11 | 52 | 52 | 52 | 52 | 11 | 49 |
| SAM_autumn | 0.32 | | | | | | | | | |
| SAM_prior | -0.06 | 0.51 | | | | | | | | |
| SAM_spring | 0.04 | 0.56 | 0.83 | | | | | | | |
| Long.E | -0.63 | -0.17 | 0.10 | 0.05 | | | | | | |
| DaysSinceSeaIce | 0.56 | 0.18 | -0.03 | 0.07 | -0.27 | | | | | |
| SST | 0.92 | 0.27 | -0.14 | -0.03 | -0.68 | 0.60 | | | | |
| Salinity | -0.43 | -0.14 | 0.31 | 0.21 | 0.23 | -0.13 | -0.41 | | | |
| year | 0.18 | 0.27 | 0.35 | 0.32 | -0.24 | 0.02 | 0.27 | -0.06 | | |
| NASA.chla | -0.02 | 0.50 | 0.72 | 0.69 | 0.11 | -0.08 | -0.15 | 0.14 | 0.43 | |
| ca | -0.15 | 0.55 | 0.57 | 0.63 | 0.20 | -0.01 | -0.20 | 0.22 | 0.13 | 0.37 |
| cc | 0.37 | 0.36 | 0.27 | 0.35 | -0.07 | 0.27 | 0.25 | -0.14 | 0.11 | 0.25 |
| cca | -0.36 | -0.02 | 0.26 | 0.20 | 0.41 | -0.12 | -0.36 | -0.07 | -0.07 | 0.20 |
| cd | 0.48 | 0.38 | 0.31 | 0.29 | -0.13 | 0.37 | 0.35 | -0.17 | 0.20 | 0.36 |
| cn | -0.70 | -0.06 | 0.42 | 0.24 | 0.48 | -0.40 | -0.69 | 0.56 | -0.04 | 0.33 |
| cyc | 0.13 | 0.09 | -0.10 | -0.03 | 0.02 | 0.32 | 0.12 | 0.02 | -0.11 | 0.03 |
| da | 0.18 | 0.37 | 0.34 | 0.27 | -0.06 | 0.18 | 0.13 | -0.08 | 0.06 | 0.37 |
| dcx | -0.57 | 0.15 | 0.06 | 0.24 | 0.52 | -0.11 | -0.57 | 0.21 | -0.15 | 0.21 |
| ds | -0.78 | -0.17 | 0.30 | 0.14 | 0.68 | -0.41 | -0.75 | 0.36 | -0.14 | 0.17 |
| dt | -0.18 | -0.44 | -0.08 | -0.16 | 0.16 | -0.19 | -0.17 | 0.23 | -0.02 | -0.10 |
| ehx | -0.28 | -0.38 | -0.42 | -0.38 | 0.21 | 0.12 | -0.25 | -0.01 | -0.37 | -0.24 |
| fcx | 0.26 | -0.06 | -0.08 | -0.09 | -0.58 | -0.08 | 0.35 | -0.12 | 0.24 | -0.15 |
| fk | 0.23 | 0.52 | 0.16 | 0.25 | -0.07 | 0.19 | 0.22 | -0.46 | -0.05 | 0.07 |
| fps | -0.13 | 0.22 | -0.02 | 0.22 | -0.10 | -0.05 | -0.03 | 0.12 | 0.22 | 0.02 |
| fr | 0.16 | -0.39 | -0.58 | -0.57 | -0.13 | 0.13 | 0.22 | -0.12 | -0.24 | -0.59 |
| fri | 0.11 | -0.10 | 0.00 | -0.03 | -0.02 | 0.02 | 0.10 | -0.03 | 0.03 | -0.01 |
| guc | 0.09 | 0.12 | -0.06 | -0.06 | 0.05 | 0.17 | 0.10 | -0.03 | -0.02 | 0.12 |
| nix | -0.47 | -0.45 | -0.29 | -0.31 | 0.42 | -0.32 | -0.46 | 0.09 | -0.22 | -0.19 |
| parm | -0.60 | -0.29 | 0.15 | -0.09 | 0.42 | -0.42 | -0.65 | 0.36 | -0.28 | 0.16 |
| pet | -0.25 | -0.13 | -0.27 | -0.08 | 0.15 | -0.17 | -0.25 | 0.02 | -0.02 | -0.04 |
| psl | -0.35 | 0.39 | 0.19 | 0.37 | 0.36 | -0.09 | -0.35 | 0.18 | 0.01 | 0.26 |
| ta | -0.16 | 0.32 | 0.12 | 0.16 | 0.15 | -0.11 | -0.11 | -0.19 | -0.15 | 0.00 |
| [NOx] | -0.77 | -0.39 | 0.23 | 0.04 | 0.53 | -0.43 | -0.72 | 0.54 | -0.14 | 0.12 |
| [PO4]  (n=51) | -0.73 | -0.56 | -0.07 | -0.26 | 0.62 | -0.52 | -0.70 | 0.39 | -0.13 | -0.10 |
| [SiO4] | -0.56 | -0.42 | 0.26 | -0.05 | 0.40 | -0.49 | -0.63 | 0.39 | 0.09 | 0.22 |
| [NOx]  (n=26: later in | -0.18 | -0.58 | -0.05 | -0.25 | -0.23 | -0.19 | 0.02 | 0.27 | -0.17 | |
| [PO4]  season | -0.13 | -0.74 | -0.51 | -0.68 | 0.09 | -0.31 | -0.01 | 0.03 | -0.02 | |
| [SiO4] samples) | -0.10 | -0.51 | -0.04 | -0.31 | -0.16 | -0.35 | -0.44 | -0.05 | 0.34 | |

**This comment is related to the last one, but we have no idea where we stand with respect to phytoplankton phenology. In Figure 1, it would be nice to have satellite-derived time series of chlorophyll a, for example. The problem I see here is that the SAM could perhaps also change the phytoplankton phenology (bloom duration or timing for example).**

- We will obtain time-series NASA satellite total chlorophyll and consider including in Figure 1

**And perhaps what the authors have defined as interannual variability driven by the SAM can simply be related to a sampling of different phenological states. It would be important for me to check this point.**

- No way to confirm this for certain, however:
- More positive SAM in the prior spring (**SAM spring**) and **SAM prior** (SAM prior to each sample) may result in the productive season commencing earlier, and thus organisms that show a decline in relative abundance through the season might show a lower relative abundance at a given time with higher **SAM spring** and **SAM prior** : 10 of the 22 taxa showed a significant correlation the time through the spring-summer of collection, of these, with 4 taxa showed a relationship with both **SAM spring** and **SAM prior** supporting the possibility that **SAM spring** and **SAM prior** were leading to an effective sampling later in the phenotypic succession (i.e. three taxa having negative relationship with sampling date and both **SAM spring** and **SAM prior**, one taxon having positive relationship with sampling date and both **SAM spring** and **SAM prior**). However, the other six taxa showing significant relationship with sampling date did not confirm this relationship.

**Specific comments: l.186-186: Can you add a table in the paper or in the supplementary materials listing these taxa (the 4 in all the samples, and the 11 in 90% of the samples)?**

- New Table will be added as Table 2:

*Table 2: Identified taxa: taxa-code, cells counted, average individual cell volume, abundance: cells/ml (average, minimum and maximum), relative abundance, total taxa-group volume (µm³/ml), relative taxa-group volume, and percentage of samples in which each taxon was identified.*

| taxon | | cells counted: number | cells measured: number | average individual cell volume (µm³) | abundance (cells/ml) average | min | max | fraction of abundance: average | average total cell volume (µm³/ml) | average fraction of total cell volume | samples with taxon |
|---|---|---|---|---|---|---|---|---|---|---|---|
| *Chaetoceros atlanticus* | ca | 589 | 479 | 1,316 | 51 | 0 | 364 | 2.2% | 81,382 | 1.4% | ***90%*** |
| *Chaetoceros castracanei* | cca | 49 | 34 | 940 | 6 | 0 | 38 | 0.3% | 18,616 | 0.4% | 48% |
| *Chaetoceros concavicornis/curvatus* | cc | 303 | 200 | 3,443 | 20 | 0 | 135 | 0.7% | 78,443 | 1.4% | 77% |
| *Chaetoceros dichaeta* | cd | 2,719 | 1943 | 491 | 423 | 0 | 2,503 | 13% | 145,999 | 2.9% | ***94%*** |
| *Chaetoceros neglectus* | cn | 650 | 488 | 176 | 83 | 0 | 697 | 3.5% | 11,906 | 0.2% | 81% |
| *Cylindrotheca closterium* | cyc | 122 | 50 | 121 | 17 | 0 | 79 | 0.7% | 4,106 | 0.1% | 77% |
| *Dactyliosolen antarcticus* | da | 748 | 472 | 61,899 | 44 | 0 | 195 | 1.6% | 1,860,680 | 27% | ***98%*** |
| *Dactyliosolen tenuijunctus* | dt | 2,121 | 1350 | 3,828 | 296 | 7 | 1,315 | 9.9% | 895,367 | 16% | ***100%*** |
| *Dictyocha speculum* (silicoflagellate) | ds | 110 | 84 | 4,920 | 10 | 0 | 69 | 0.5% | 99,301 | 1.5% | 48% |
| discoid centric diatoms | dcx | 1,280 | 1280 | 8,572 | 133 | 12 | 696 | 5.2% | 437,556 | 7.3% | ***100%*** |
| *Emiliania huxleyi* (haptophyte) | coc | 173 | 50 | add | 24 | 0 | 192 | 0.8% | 3,552 | 0.1% | 58% |
| *Fragilariopsis cylindrus/curta* | fcx | 3,987 | 3013 | 70 | 632 | 0 | 8,796 | 17% | 44,167 | 0.9% | ***98%*** |
| *Fragilariopsis kerguelensis* | fk | 4,428 | 4055 | 3,748 | 167 | 0 | 1,054 | 5.8% | 369,492 | 6.5% | ***98%*** |
| *Fragilariopsis pseudonana* | fps | 170 | 115 | 355 | 26 | 0 | 201 | 0.9% | 18,999 | 0.4% | 69% |
| *Fragilariopsis rhombica* | fr | 4,542 | 3469 | 36 | 658 | 29 | 2,070 | 22% | 23,359 | 0.6% | ***100%*** |
| *Fragilariopsis ritscheri* | fri | 46 | 19 | 572 | 7 | 0 | 86 | 0.2% | 11,020 | 0.2% | 35% |
| *Guinardia cylindrus* | guc | 119 | 81 | 10,405 | 15 | 0 | 79 | 0.6% | 225,921 | 4.1% | 67% |
| *Nitzschia acicularis/decipiens* | nix | 1,133 | 509 | 251 | 162 | 0 | 977 | 5.7% | 46,705 | 1.0% | ***98%*** |
| *Parmales spp.* (chrysophyte) | parm | 322 | 2 | 8 | 38 | 0 | 668 | 1.7% | 334 | 0.0% | 27% |
| *Petasaria heterolepis* (other) | pet | 45 | | | 7 | 0 | 187 | 0.3% | 2,667 | 0.1% | 6% |
| *Pseudonitzschia lineola* | psl | 703 | 403 | 1,093 | 91 | 4 | 376 | 4.1% | 84,460 | 1.5% | ***100%*** |
| *Thalassiothrix antarctica* | ta | 287 | 269 | (63,000) | 13 | 0 | 172 | 0.6% | 314,424 | 4.8% | 85% |

**Table: Table 1: Long.E is indicated two times as variable, is it an error?**

- Yes, a typing error – the second occurrence of Longitude in Table 1 should be Latitude

---

## Author Comment (AC2) · 23 Dec 2019

Response to comments on the submitted manuscript: Greaves et al. - *The Southern Annular Mode (SAM) influences phytoplankton communities in the seasonal ice zone of the Southern Ocean*

23 December 2019
* * *
We thank the reviewer for their valuable feedback on this manuscript. These have identified several areas for improvement of the manuscript, which we have addressed below:

**RC2 - Damiano Righettim, 21 November 2019**

**(i) Key concepts (SAM or SAM index) are not clearly defined. The SAM definition leaves it open to the reader, how the sign of the SAM index is calculated, and whether atmospheric pressure or water pressure constitutes the SAM index.**

- SAM is calculated by NOAA (USA), as already stated in the Methods section: "*Daily estimates of SAM were obtained from the US NWS Climate Prediction Center's website and are the NOAA Antarctic Oscillation Index values based on 700-hPa geopotential height anomalies (NOAA, 2017)*." (line 140) – the description wording is as specified by NOAA. We will clarify the definition of SAM and the SAM index (line 58 onwards): "*The Southern Annular Mode (SAM, also called the High-Latitude Mode or the Antarctic Oscillation) is the principal mode of atmospheric variability over the Southern Ocean (Gong andWang, 1999; Marshall, 2003), characterised by large-scale movement of air mass between high and mid latitudes. Importantly for this study, it determines the strength and latitudinal variation of the westerly wind belt. There are various definitions and indices for SAM (Ho et al, 2012) - it has been defined as the difference in normalised zonal mean atmospheric sea-level pressure between 40°S and 65°S (Gong and Wang, 1999; Marshall, 2003), however we have used the NOAA Antarctic Oscillation Index values, which are based on 700-hPa geopotential height anomalies (NOAA, 2017). More positive values of the SAM index lead to stronger westerly winds in high latitudes, including the study area (Hall and Visbeck 2002; Lovenduski and Gruber 2005; Arblaster and Meehl, 2006)*".
- The SAM is an atmospheric index, we will clarify by adding the word "*atmospheric*" to the Introduction paragraph of section 1.2 (as underlined here): "*The Southern Annular Mode (SAM), which is also variously also called the High-Latitude Mode and the Antarctic Oscillation, has been defined as the difference in normalised zonal mean atmospheric sea-level pressure between 40°S and 65°S (Gong andWang, 1999; Marshall, 2003)*." (line 58)
- We will include "*SAM index (hereafter referred to as SAM)*" in line 58 to clarify any confusion, and include the previously observed range in SAM (in the order of "*-3 to +3*") in the text to improve clarity.

**There is a problem with clarity of statements and consistency of word use (e.g., different expressions are used for the same thing), and a lack of clear correspondence between hypothesis, methods, and key results. I provide detailed examples on clarity below.**

- These will be checked for consistency

**(ii) My main conceptual critique point is that the impact of the time-averaged SAM signal in autumn on phytoplankton community composition in spring to summer has not been firmly tested by the data shown.**

- The relationship between time-averaged SAM signal in autumn on phytoplankton community composition was apparent in the analysis, and reasonable (being the time ice was forming) but otherwise untestable. However:
- Correlations with the empirically defined SAM range in the autumn and the relative abundances of 12 of the 22 taxa supported the conclusion. Further:
- The peak of SAM influence in the preceding autumn was also detected in response surfaces for NASA satellite total chlorophyll (correlation between SAM in autumn and NASA total chlorophyll is 0.5) and nutrient levels (correlation between SAM in autumn and $[PO_4]$ was -0.64 for all samples, and -0.84 for the later-in-season half of the samples) – these response surfaces will be included in the extra material (as drafted below). NASA satellite total chlorophyll and $[PO_4]$ are observationally independent of the taxonomic counts, so similar prior-autumn maxima for the correlation with SAM and these traits are supportive of our finding that "*time-averaged SAM signal in autumn influences phytoplankton community composition in spring to summer*"
- Two Supplementary material figures to be included:

[Figure]

*Figure [Supplementary material]: Response surfaces of the correlation between $[PO_4]$ and the averaged SAM, versus timing and length of the daily-SAM averaging range, i.e. the calendar date of the mid-point of the date range (horizontal axis), and the number of days over which those indices were averaged (vertical axis), respectively. The SAM maxima identified in Figure 3 are shown (SAM autumn, SAM spring). Evident maxima in autumn are indicated with red broken line loops. (a) Analysis includes all available data (n=51), (b) analysis includes only half of the samples, being those collected later in the spring-summer productive season (n=26).*

[Figure]

*Figure [Supplementary material]: Response surfaces of the correlation between NASA satellite total chlorophyll and the averaged SAM, versus timing and length of the SAM period. The SAM period is the number of days of daily-SAM averaged (vertical axis) and the timing of the range of averaged daily-SAM (horizontal axis). The SAM maxima identified in Figure 3 are shown (SAM autumn, SAM spring). Evident maxima in autumn are indicated with red broken line loops. (a) Analysis includes all available data (n=51), (b) analysis includes only half of the samples, being those collected later in the spring-summer productive season (n=26).*

**The study demonstrates that it is possible to average the daily SAM index in a way that a significant part of the variation in community composition can be explained in next spring/summer, yet it is unclear why microbial species that live on timescales from days to weeks, would respond to the SAM signal with a time-lag of several months.**

- We will include in the discussion "*Phytoplankton taxa must survive the six months of darkness and ice-cover between the middle of the Austral autumn and mid-spring by variously hibernating and/or producing resting spores (see manuscript, as described from line 290) so their metabolic condition in autumn is likely to determine their viability and relative vigour in spring.*"

**I suggest that relationships between a more positive state of SAM in autumn and temperature, wind speed, mixed-layer depths, and nutrient levels in spring to summer—factors that may directly shape phytoplankton composition—shall be evaluated, to support the paper's message.**

- "**relationships between a more positive state of SAM in autumn and temperature, wind speed, mixed-layer depths, and nutrient levels in spring to summer**" are beyond the scope of this paper – others have made observations/predictions of the influence of the SAM on wind-speed and mixed-layer-depths as cited: "*More positive SAM has been associated with lower atmospheric pressure at sea level and increased storminess (Kwok and Comiso, 2002; Hall and Visbeck, 2002; Marshall, 2007). These changes are particularly marked south of 60°S in the atmospheric Southern Circumpolar Trough (Hines et al., 2000; Mackintosh et al., 2017), a region characterised by strong winds with variable direction (Taljaard, 1967). Stronger winds may result in increased transport of surface water northward from the*

*Antarctic Divergence by Ekman drift (Lovenduski and Gruber, 2005; DiFiore et al., 2006), potentially driving increased upwelling of nutrient- and carbon-rich deep ocean water at the Antarctic Divergence (Hall and Visbeck, 2002). More positive SAM is also associated with reduced near-surface air temperature over the SIZ due to an increased frequency of strong southerly winds and increased cloud cover (Lefebvre et al., 2004; Sen Gupta and England, 2006).*"

- We don't have wind-speed and mixed-layer-depth for each sample, and we would require this information daily for the location of each sample for the year preceding each sample, and arguably for a range of locations around each sample (since the surface water is migrating horizontally at speeds up to 15 cm s$^{-1}$, as discussed from line 397). Whilst this would be an interesting analysis, it is far beyond the scope of this paper, which was to determine if an effect of the SAM could be detected in the taxonomic composition of phytoplankton (as stated in our hypothesis)
- Nutrients are replete at the start of spring, assumed to be unaffected by SAM the previous autumn
- It is possible that SAM in the autumn influences SAM in the following spring – we did not identify a significant correlation between SAM in the autumn and SAM in the following spring (Table 2 of the submitted manuscript)

**In section 4 ('Other relationships'), there are several relationships presented between predictors, yet the results are not presented in a structured way to support the hypothesis that SAM-induced changes in temperature, wind-speed, mixed-layer depth or nutrient concentrations affect community composition.**

- These other relationships are worthy of inclusion – some introductory text will be added to the section to make this more clear
- We did not hypothesise that "**SAM-induced changes in temperature, wind-speed, mixed-layer depth or nutrient concentrations affect community composition**", our hypothesis was: "*Based on the predicted and observed positive relationships between the SAM and phytoplankton productivity and biomass in the SIZ of the SO, we hypothesised that changes in the SAM could also elicit changes in the composition and abundance of the phytoplankton community.*" (line 93)

**The current association between the SAM signal (or "SAM modes") described and community composition may not be causal. In the context of fast-lived organisms it seems crucial to test if the link between summer community composition and (preceding) SAM is plausible.**

- Sure, may not be causal – but it is plausible, as discussed above. Without conducting a series of overwintering experiments, there is no way to check for sure.
- We would expect SAM prior to sampling (**SAM prior** and **SAM spring**) would influence phytoplankton composition, as we would expect SAM in the winter to have a lesser influence because the surface-ocean is insulated from the atmospheric conditions by sea-ice. These expected influences were observed (Figure 3). The influence of SAM the previous autumn was not expected, but is considered a real influence as it is the time when sea-ice is forming and thus a critical time for phytoplankton preparing to hibernate the six-months of ice-cover.
- Further, the empirically defined SAM autumn showed pairwise correlations with 12 of the 22 taxa identified.

- Further, SAM maxima were apparent in similar response surface analysis of the correlation between SAM and (a) NASA satellite total chlorophyll, and (b) $[PO_4]$ in all samples, and (c) as a stronger correlation with $[PO_4]$ when only the later-in-the-season half of samples were considered (analysis not included in original manuscript, but now to be included with response surface figures in Supplementary Material – as indicated above)

**Recommendations**

**I suggest that the manuscript is thoroughly screened for clarity.**

- We will carefully review and improve the manuscript for clarity

**Second, besides further testing the associations of the SAM signal of autumn with physicochemical factors known to affect phytoplankton composition (and whether these associations are in line with expectation), I suggest splitting the 22 taxa into ecological test groups, which are expected to respond differently to changing mixing-, wind-, and nutrient patterns under a more positive SAM state. These expectations can be presented as specific hypotheses in the introduction.**

- We do not believe that we have enough information about enough of the identified taxa to be able to sensibly break the identified taxa into groups that will lead to a sensible group-based analysis of responses to SAM. Not a great deal is known about many identified Southern Ocean polar hard-shelled phytoplankton Supplementary taxa, which have previously largely been to only identified at the genera level, and we have identified significant differences in the behaviour of taxa within single genera.

**Such a biological approach has been partly implemented by comparing small diatoms (presumably better adapted to stable waters) with large diatoms (presumably better adapted to strong mixing). Yet the results of this test lack a graphical presentation in the manuscript, across all taxa.**

- We will interpret the observed results with reference to organism size and shape to infer any influence
- However, size is not necessarily a useful parameter upon which to aggregate taxa, as whilst some taxa are always small, others have been identified as both large and small taxa.

**Species may be grouped further into warm, temperate, or polar species, depending on their global distributions (e.g. using observations from OBIS and GIBF; Righetti et al., 2019) and their responses may differ under SAM-induced warming/cooling.**

- We will endeavour to associate "warm, temperate, or polar" to each taxon after OBIS and GIBF; Righetti et al., (2019) to ascertain if such grouping supports the observed taxa responses to the environmental variables

**Similarly, R-strategist (fast growing, light stress tolerant species) and S-strategists (slow growing, nutrient stress tolerant species) may be grouped together (Brun et al., 2015), as they may respond oppositely to changing nutrient levels.**

- Brun et al (2015) reproduces the *R-S-strategist* classification of organisms from Reynolds (2006): of the 22 taxa in our study, only 4 were classified (as *R-strategists*), and 5 were classified as "*unclassified*", with 15 not included in Brun et al (2015)'s reproduction of Reynolds (2006) classification [we haven't as yet been able to access Reynolds, C.S., 2006. *The ecology of phytoplankton*. Book: Cambridge University Press]. Given the paucity of *R-S-*

*strategist* classifications (4 out 22 taxa with classifications) it would be inappropriate to specifically overlay the *R-S-strategist* framework on the taxonomic data we have collected.

- We propose to include discussion of the *R-S-strategist* classification in our re-worked discussion.

**Additionally, species with large vs. small cells may show opposite responses to changing turbulence and wind regimes (Margalef, 1997, 1978).**

- We propose to include in our re-worked discussion more discussion of size and shape (after Margalef 1997) of the taxa that showed influence of SAM

**Finally, predicting the response of siliceous vs. calcareous taxa to SAM constitutes an exciting hypothesis: these groups have shown opposite responses to deeper mixing or nutrient entrainment (Cermeño et al., 2008).**

- The area studied was the Seasonal Ice Zone (SIZ) which is situated over the ocean upwelling zone of the Antarctic Divergence – nutrients in the surface waters of the SIZ are replenished over the six months when the sea surface is ice-covered and when there is almost no productivity (or consumption of nutrients). It is considered a high-nutrient, low-chlorophyll zone. In this region of annual winter nutrient replenishment, the influence of mixed-layer depth is less than most other areas of the world's ocean. The area falls outside the analysis of Cermeño et al., (2008), whose sampled area extended southward to only well north of the Antarctic Divergence, and would not be expected to conform to the trends observed by Cermeño et al., (2008) due to the replenishment of nutrients every winter in the area of our study.

**With respect to the clustering techniques used to describe communities I cannot give detailed recommendations, as the metrics used are beyond my expertise.**

**Detailed comments**

**There are too many comments to be listed. I therefore give examples for selected paragraphs, with comments on clarity, for each:**

**Abstract:**

**- Line 3: How many variables were tested?**

- We will amend the text to include

**- Line 6: How many species (genera, higher taxa) were included among the 22 taxa?**

- We will amend the text to include

**- Line 7: I do not understand 'CAP'. This term has not been introduced.**

- We will amend the text to include the full name of the CAP procedure

**- Lines 8, 9, 11, 17: The following terms are used: taxonomic community composition, taxa composition, phytoplankton community structure, taxonomic composition of phytoplankton. While I understand that the authors strive to include stylistic variation, the reader is confused by the multiple expressions. Do they denote the same thing or not? I recommend using use the same expression for the same thing. Else, once an expression is clear, an abbreviation of the latter may be used therein, as long as it denotes the same thing.**

- We will amend the text to make the terms consistent

**- Line 10: Unclear to me, if the correlation is significant or not.**

- We will amend the text to clarify

**- Line 13: Unclear to me, if "response" means a response of abundance or not.**

- We will amend the text to clarify

**- Line 15: Before, the expression "SAM index" was used, not "higher SAM". Does "higher SAM" refer to a more positive state of the SAM index?**

- We will amend the text to clarify

**- Line 17: Confusing, as taxonomic composition of phytoplankton is not the same thing as a standing stock (or a "pasture") of biomass of phytoplankton.**

- We will amend the text to clarify

**- Line 16: It is unclear to me, if the expression "pelagic ecosystem" is suitable in the context of a sea ice transition zone.**

- We will remove the word "*pelagic*" to avoid confusion

**- Line 16: It is unclear, how many of the total species that were studied, responded significantly to SAM. Thus, it is unclear, if this result is important or general.**

- We will amend the text to clarify

**- Line 10 ff: It is surprising that 'day of sampling' explains more variation in community composition than any other locally sampled environmental factor (SST, nutrients, etc). An interpretation on why this is the case would help the reader to assess the plausibility or importance of this result.**

- The seasonal ice zone has been previously observed and reported to have a winter period (around 6 months) with little or no phytoplankton productivity when the sea-surface is frozen, and a well-characterised bloom and systematic taxonomic succession through the spring-summer months as sea-ice melts – we will include this point in the abstract (it is already in the introduction and in the discussion)

**Introduction:**

**- Line 21-23: The first two sentences are partially repetitive.**

- We will improve the text

**- Line 21 ff: The paragraph wants to establish the importance of phytoplankton productivity in the study area for global phytoplankton productivity. While the reader understands that a larger fraction (~30%) of carbon fixed by phytoplankton is exported in the study region, relative to the global average (~ 20% exported) it remains unclear, if the study region is globally important. What is the area-weighted contribution of the study region to global phytoplankton C-export?**

- We will include statement to this effect indicating the fraction of global C-export attributed to the SIZ of the Southern Ocean.

**1.2 The Southern Annular Mode**

**- Line 58 ff: Clarify the definition of SAM (see above). The reader cannot grasp how the sign of SAM is calculated or linked to changing pressure gradients, and thus how it is associated with physicochemical changes in the study system.**

- The calculation of SAM is beyond the scope of this paper, except in the most general terms – the SAM index used was calculated by NOAA (USA) and the wording of the description is as NOAA wants it reported.
- We will endeavour to craft a further, greatly simplified, single-sentence description of the SAM for inclusion around line 58.

**- Line 64 ff: SAM vs. SAM index vs. SAM state vs. SAM mode. Please use consistent expressions throughout the manuscript. …… and the use of "mode" in both the context of SAM and community composition may confuse the reader**

- We will remove all references to SAM maxima as MODES to remove confusion with MODE=MAXIMA and SOUTHERN ANNULAR MODE, that is, we will just define the terms ***SAM spring*** and ***SAM autumn*** and refer to them by name without using the term "*mode*".

**In addition, "taxon" could always refer to both a species and a group of species.**

- *Taxon* is singular, *taxa* is plural – we have used "*taxon*" to refer to a single taxa-group, whether it is a single taxon or a group of taxa, and we have used "*taxa*" – we will carefully consider each usage of "*taxa*" and "*taxon*" for consistency

**2.1 Phytoplankton composition and abundance**

**- Line 116: One reads as if the abundance of phytoplankton communities was sampled. As much as I understand, the abundance of species or taxa was sampled. (Then, an abundance-weighted community composition was calculated?).**

- Improve text to remove any suggestion that "**abundance of phytoplankton communities was sampled**"

**2.3 Statistical analysis**

**- Line 151 ff: The methods section needs clarification, structurally and through editing. In this section, I have difficulties to understand whether three or more sets of analyses were performed based on the phytoplankton field data, and which of these analyses is most important to test the key hypothesis of the paper,**

- We will edit methods to improve clarity

**….. and at what temporal resolution the analyses were performed.**

- Temporal resolution is specified (line 147-): "*These were derived by evaluating separate CAP analyses (described below) based on daily SAM averaged across a range of days {1, 3, 5, . . . 365 days} centred on (i) each calendar day individually (1 Jan – 31 Dec) through the year associated with each sample; and (ii) lagged from 1 to 365 days prior to each sample collection date.*" – We consider this description to be succinct, but we will craft a more descriptive and wordy description for readers who may be unable to interpret the existing description.

- We will add *"Heat maps were used to display the variance explained from individual CAP analyses according to the number of days and mid-point (or lagged mid-point) used for each derived SAM average, thereby allowing an investigation of the derived SAM index showing greatest correlation with community structure."*

**- Line 152: Has "community structure" really been correlated to "environmental covariates"? If I understand correctly, the abundance data was related to possible environmental drivers, per species. In this case, please specify: e.g. …and species abundance between samples**

- The correlation between the community structure (as determined from the ordination) and each environmental covariate was calculated according the procedure outlined in ter Braak (1995) and attributed to Dargie (1984) – we will include more explanation of CAP analysis for greater ease of interpretation for readers not familiar with CAP analysis.

**- Line 151 ff: It is not clearly motivated, why clustering of community-level samples is suitable to identify the effect of SAM on community composition. To me, the number of 52 samples seems rather low already, and each degree of freedom may be valuable.**

- Clustering shows that there are significant differences between the community composition of the samples. Clustering does not identify an effect of SAM, at least not directly, since environmental covariates are not included in the cluster analysis. The group structure determined by cluster analysis is displayed in the CAP ordination (using colour) to demonstrate that samples that clustered together are indeed close to one another in the 2D ordination. This lends confidence that the 2D ordination is a reasonable approximation to the full, high-dimensional structure. As we know the values for the environmental covariates for each sample, it is possible to determine the correlation between the 2D CAP solution and each environmental covariate. We display this correlation as a projected vector (arrow) where direction indicates the sign and length indicates strength. For example, Figure 6 shows that group 5 (dark blue points) are positively correlated with Long.E, but negatively correlated with DaysAfter1Oct (at least up to the approximation afforded by the first two canonical axes).

**3. Results**

**- The first results presented to the reader are abundance-distributions of taxa across samples. Yet, the reader might expect that the most important piece of evidence to elucidate the role of SAM for phytoplankton composition is first presented.**

- The logic behind the analysis and presentation of the data in relation to the hypothesis will be explained in the first paragraph of the *Results* section
- We will re-arrange the *Results* section to put abundance information later, i.e. Section 3.1 "*Observed abundance*" will become Section 3.2, with Section 3.1 becoming "*CAP analysis and pair-wise relationships*"

**- Line 206 ff. Can P, n, and R2-values be provided for the correlations?**

- The text will be modified to include specification of n and p with all in-text correlations
- P-values are not tabulated - currently reported correlations in tables are formatted to indicate when p<0.05 and when p<0.05/20 for Bonferroni correction

**- Table 1: I do not understand, why nutrients are excluded in this table.**

- We will include text to explain that nutrient levels are an effect of phytoplankton, not a cause, in this high-nutrient-low-chlorophyll environment that is nutrient-replenished through the winter – i.e. more growth through the productive spring-summer leads to less nutrition at the end of the summer

**- Figure 5. The caption remains vague. What are the "several underlying assumptions" of linear regression? Relevant to be discussed in the caption?**

- We agree that the caption should be improved and we will remove the words "*though clearly several underlying assumptions of linear regression would not be met*" from the caption

**Overall, the manuscript requires a clear structure in order to show to what degree the SAM signals may matter to community composition, based on (ecological) hypotheses tested and data. The support in the data for this message, and the evaluation of the manuscript are complicated at current and warrant further attention.**

- We will reconsider the structure for the manuscript.

---

## Author Response (AR1)

Response to comments on the submitted manuscript: Greaves et al. - *The Southern Annular Mode (SAM) influences phytoplankton communities in the seasonal ice zone of the Southern Ocean*

13 February 2019
* * *
We thank the reviewer for their valuable feedback on this manuscript. These have identified several areas for improvement of the manuscript, which we have addressed below:

**RC1 - Anonymous Referee #1, 16 November 2019**

**In this manuscript entitled "The Southern Annular Mode (SAM) influences phytoplankton communities in the seasonal ice zone of the Southern Ocean", the authors examine the role of SAM on phytoplankton communities in the SIZ of the Southern Ocean.**

**I think the document is not yet ready to be published, although the subject and results are really interesting.**

- Certainly, the results are very interesting

**The structure of the document is really difficult to follow at the moment.**

- We have carefully reviewed and improved the structure of the manuscript in reference to the comments of both reviewers

**I have listed some improvements that could be made to improve the clarity of the manuscript.**

**General comments: My main concern is related to the structure of the document, to many subsections, particularly in the sections on results and discussion (8 subsections for discussion, and 2 sentences for conclusion, 1 sentence in the section on results (3.1). The document, as it is now, is unbalanced and difficult to read and needs to be reorganized around major themes (seasonal, interannual variability and impact on phytoplankton communities for example for the discussion).**

- We have refined the manuscript structure, which we consider will fix this issue

**In this paper, the authors examined the role of SAM and seasonal variability on changes in phytoplankton communities, but some key environmental factors are really missing in this study, (1) mixing estimates (by estimating the depth of mixed layers, deriving wind stress)**

- We don't have this information for each sample, or for the time periods prior – we are surmising that SAM influences wind-speed and subsequently mixed-layer-depth from the previously published observed and predicted positive relationship between the SAM and wind speed.
- While we are unable to estimate the correlations the referee asks for it is behest upon us to ensure that these factors are included in the manuscript, the variance from which probably contributes to the 62.5% of unexplained residual variance in this study

**and (2) light measurements (in situ or satellite data)?**

- We don't have this information for each sample

Because it can be suspected that changes in the intensity of the SAM will directly influence light-mixing regimes, and therefore changes in the composition of phytoplankton communities at the time of sampling?

This is particularly important given that the authors mention the interaction between mixing and phytoplankton dynamics in the discussion.

- It has been previously reported that SAM has been observed and predicted to relate to wind intensity (from line 83) – thus we used this to help explain how the identified maxima in SAM relationship with phytoplankton taxonomic composition could be plausible (from line 329)

In addition, the authors focused on understanding changes in the relative abundance of the main phytoplankton groups, but we have no idea how phytoplankton biomass could change annually with the SAM.

- Previous researchers have concluded that long term changes in the SAM will influence productivity: "*Lovenduski and Gruber (2005) predicted that increased SAM would support higher phytoplankton productivity, and subsequent analyses by Arrigo et 90 al. (2008); Boyce et al. (2010), and Soppa et al. (2016) have confirmed a positive relationship between the SAM and phytoplankton standing stocks and productivity south of 60°S in the SIZ*" (from line 88)
- We have now included satellite-derived estimates of Total Chlorophyll as an index of biomass which we had been able to obtain for 49 of the 52 samples, which also show a positive relationship with autumn SAM, i.e. higher SAM in autumn is associated with higher NASA satellite total chlorophyll in the following spring-summer (Table 2, reported in Results from line 294)
- The peak of SAM influence in the preceding autumn was also detected in response surfaces for NASA satellite total chlorophyll (correlation between SAM in autumn and NASA total chlorophyll is 0.5) and nutrient levels (correlation between SAM in autumn and [$PO_4$] was -0.64 for all samples, and -0.84 for the later-in-season half of the samples) – these response surfaces will be included in the extra material (as drafted below). NASA satellite total chlorophyll and [$PO_4$] are observationally independent of the taxonomic counts, so similar prior-autumn maxima for the correlation with SAM and these traits are supportive of our finding that "*time-averaged SAM signal in autumn influences phytoplankton community composition in spring to summer*"

The authors mentioned this briefly in the discussion (5.3), but can you access to any vertically integrated biomass proxies (vertically integrated chlorophyll, PP and satellite-derived estimates)?

- We inferred a productivity effect of SAM from nutrient drawdown, which showed reduced nutrients with more positive prior SAM indices, with the relationships with prior SAM indices (*SAM spring*, *SAM prior*, and *SAM autumn*) all being stronger when only the samples collected later in the season (the later half of samples) were included. In the SIZ of the Southern Ocean, surface-water nutrition is replenished through the winter by upwelling of deep ocean water at the Antarctic Divergence. The nutrient contents later in the spring-summer better reflect the total production over the spring-summer than do all samples, including those collected earlier in the spring-summer (as tabulated in the new Table 2). We have included the response surfaces for the correlation between the SAM and [$PO_4$] in Supplementary Material (Fig. S1). This point is now discussed (from line 370)

- We have added NASA satellite total chlorophyll estimates which we had been able to obtain for 49 of the 52 samples, which also show a positive relationship with SAM, i.e. higher SAM is associated with higher NASA satellite total chlorophyll (now in Table 2)

**This comment is related to the last one, but we have no idea where we stand with respect to phytoplankton phenology. In Figure 1, it would be nice to have satellite-derived time series of chlorophyll a, for example. The problem I see here is that the SAM could perhaps also change the phytoplankton phenology (bloom duration or timing for example).**

- We have added time-series NASA satellite total chlorophyll to Figure 1

**And perhaps what the authors have defined as interannual variability driven by the SAM can simply be related to a sampling of different phenological states. It would be important for me to check this point.**

- No way to confirm this for certain, however:
- The CAP analysis fits multiple covariates, apportioning variance to each – the variance in community composition explained by the time through the spring-summer that a sample was collected (***DaysAfter1Oct***) was orthogonal to the variance explained by the SAM, and thus we conclude the apparent variance associated with the SAM to be independent of the variance associated with sampling "**different phenological states**" associated with the time through the spring-summer.
- More positive SAM in the prior spring (***SAM spring***) and ***SAM prior*** (SAM prior to each sample) may result in the productive season commencing earlier, and thus organisms that show a decline in relative abundance through the season might show a lower relative abundance at a given time with higher ***SAM spring*** and ***SAM prior*** : 10 of the 22 taxa showed a significant correlation the time through the spring-summer of collection, of these, with 4 taxa showed a relationship with both ***SAM spring*** and ***SAM prior*** supporting the possibility that ***SAM spring*** and ***SAM prior*** were leading to an effective sampling later in the phenotypic succession (i.e. three taxa having negative relationship with sampling date and both ***SAM spring*** and ***SAM prior***, one taxon having positive relationship with sampling date and both ***SAM spring*** and ***SAM prior***). However, the other six taxa showing significant relationship with sampling date did not confirm this relationship.

**Specific comments: l.186-186: Can you add a table in the paper or in the supplementary materials listing these taxa (the 4 in all the samples, and the 11 in 90% of the samples)?**

- New Table has been added as Table 3, listing taxa

**Table: Table 1: Long.E is indicated two times as variable, is it an error?**

- Yes, a typing error – the second occurrence of Longitude in Table 1 should have been Latitude – this is now corrected

**RC2 - Damiano Righettim, 21 November 2019**

**(i) Key concepts (SAM or SAM index) are not clearly defined. The SAM definition leaves it open to the reader, how the sign of the SAM index is calculated, and whether atmospheric pressure or water pressure constitutes the SAM index.**

- We have edited the text and added more simplistic description of the SAM: in the Introduction from line 66, and in the Methods line 160
- The SAM is an atmospheric index, we will clarify by adding the word "*atmospheric*" to the Introduction paragraph of section 1.2 (line 66)

**There is a problem with clarity of statements and consistency of word use (e.g., different expressions are used for the same thing), and a lack of clear correspondence between hypothesis, methods, and key results. I provide detailed examples on clarity below.**

- These have been edited to correct inconsistencies

**(ii) My main conceptual critique point is that the impact of the time-averaged SAM signal in autumn on phytoplankton community composition in spring to summer has not been firmly tested by the data shown.**

- The relationship between time-averaged SAM signal in autumn on phytoplankton community composition was apparent in the analysis, and reasonable (being the time ice was forming) but otherwise untestable. However:
- Correlations with the empirically defined SAM range in the autumn and the relative abundances of 12 of the 22 taxa supported the conclusion. Further:
- The peak of SAM influence in the preceding autumn was also detected in response surfaces for NASA satellite total chlorophyll (correlation between SAM in autumn and NASA total chlorophyll is 0.5) and nutrient levels (correlation between SAM in autumn and [$PO_4$] was -0.64 for all samples, and -0.84 for samples collected in the latter half of the season) – these response surfaces have been included in the Supplementary Material (Figs. Si and S2). NASA satellite total chlorophyll and [$PO_4$] are observationally independent of the taxonomic counts. Detecting similar prior-autumn maxima for the correlation with SAM and these traits are supportive of our finding that "*time-averaged SAM signal in autumn influences phytoplankton community composition in spring to summer*" and the discussion has been improved to reflect this (Results line 293, Discussion line 370)

**The study demonstrates that it is possible to average the daily SAM index in a way that a significant part of the variation in community composition can be explained in next spring/summer, yet it is unclear why microbial species that live on timescales from days to weeks, would respond to the SAM signal with a time-lag of several months.**

- We have included more detail in the discussion to address this issue (Discussion from line 290)

**I suggest that relationships between a more positive state of SAM in autumn and temperature, wind speed, mixed-layer depths, and nutrient levels in spring to summer—factors that may directly shape phytoplankton composition—shall be evaluated, to support the paper's message.**

- "**relationships between a more positive state of SAM in autumn and temperature, wind speed, mixed-layer depths, and nutrient levels in spring to summer**" are beyond the scope of this paper – The SAM is already a proxy for such variables as wind speed, mixed depth (and thus light availability), nutrient upwelling, all of which are regarded as primary determinants of phytoplankton community structure. Others have made observations/predictions of the influence of the SAM on wind-speed and mixed-layer-depths as cited (from line 82).

- We don't have wind-speed and mixed-layer-depth for each sample, and we would require this information daily for the location of each sample for the year preceding each sample, and arguably for a range of locations around each sample. Whilst this would be an interesting analysis, it is far beyond the scope of this paper, which was to determine if an effect of the SAM could be detected in the taxonomic composition of phytoplankton (as stated in our hypothesis).
- It is possible that SAM in the autumn influences SAM in the following spring – we did not identify a significant correlation between SAM in the autumn and SAM in the following spring (Table 2 of the submitted manuscript)

**In section 4 ('Other relationships'), there are several relationships presented between predictors, yet the results are not presented in a structured way to support the hypothesis that SAM-induced changes in temperature, wind-speed, mixed-layer depth or nutrient concentrations affect community composition.**

- We have removed the section 'Other relationships' and included relevant findings from this section elsewhere in the Results as appropriate.
- We did not hypothesise that "**SAM-induced changes in temperature, wind-speed, mixed-layer depth or nutrient concentrations affect community composition**", our hypothesis was: "*Based on the predicted and observed positive relationships between the SAM and phytoplankton productivity and biomass in the SIZ of the SO, we hypothesised that changes in the SAM could also elicit changes in the composition and abundance of the phytoplankton community.*" (line 93)

**The current association between the SAM signal (or "SAM modes") described and community composition may not be causal. In the context of fast-lived organisms it seems crucial to test if the link between summer community composition and (preceding) SAM is plausible.**

- Sure, may not be causal – but it is explicable, as discussed from line 328. Without conducting a series of overwintering experiments, there is no way to check for sure.
- Further, the empirically defined SAM autumn showed pairwise correlations with 12 of the 22 taxa identified.
- Further, SAM maxima were apparent in similar response surface analysis of the correlation between SAM and (a) NASA satellite total chlorophyll, and (b) $[PO_4]$ in all samples, and (c) as a stronger correlation with $[PO_4]$ when only the later-in-the-season half of samples were considered (analysis not included in original manuscript, but now to be included with response surface figures in Supplementary Material – as indicated above)

**Recommendations**

**I suggest that the manuscript is thoroughly screened for clarity.**

- We have carefully reviewed and improved the manuscript to improve the clarity, readability and pertinence of the text, including the removal of extraneous words

**Second, besides further testing the associations of the SAM signal of autumn with physicochemical factors known to affect phytoplankton composition (and whether these associations are in line with expectation), I suggest splitting the 22 taxa into ecological test groups, which are expected to**

**respond differently to changing mixing-, wind-, and nutrient patterns under a more positive SAM state. These expectations can be presented as specific hypotheses in the introduction.**

- With respect, the niches of Antarctic phytoplankton are not sufficiently well known to identify ecological test groups of phytoplankton that are expected to respond differently to the environmental changes wrought by SAM. Instead and importantly, for the first time this paper identifies indicator species for the effects of SAM: winners and losers under increasingly positive values of SAM.
- We do not believe that we have enough information about enough of the identified taxa to be able to sensibly break the identified taxa into groups that will lead to a sensible group-based analysis of responses to SAM. Not a great deal is known about many identified Southern Ocean polar hard-shelled phytoplankton taxa, which have previously largely been only identified at the genera level, and we have identified significant differences in the behaviour of taxa within single genera.
- We have included discussion around this point from line 389.

**Such a biological approach has been partly implemented by comparing small diatoms (presumably better adapted to stable waters) with large diatoms (presumably better adapted to strong mixing). Yet the results of this test lack a graphical presentation in the manuscript, across all taxa.**

- We will interpret the observed results with reference to organism size and shape to infer any influence
- However, size is not necessarily a useful parameter upon which to aggregate taxa, as whilst some taxa are always small, others have been identified as both large and small taxa.

**Species may be grouped further into warm, temperate, or polar species, depending on their global distributions (e.g. using observations from OBIS and GIBF; Righetti et al., 2019) and their responses may differ under SAM-induced warming/cooling.**

- This is implausible. The species are almost exclusively Antarctic in geographic range. The Polar Front is a very effective barrier to the transmission of phytoplankton from warmer waters and, as above, their niches are poorly known. Only 10 of the 22 taxa/taxa-groups considered in our research had data-records in OBIS (table below) – not enough to meaningfully group the taxa for analysis – we have included Discussion around this point from line 389.

**Similarly, R-strategist (fast growing, light stress tolerant species) and S-strategists (slow growing, nutrient stress tolerant species) may be grouped together (Brun et al., 2015), as they may respond oppositely to changing nutrient levels.**

- Brun et al (2015) reproduces the *R-S-strategist* classification of organisms from Reynolds (2006): of the 22 taxa/taxa-groups in our study, only 4 were classified (as *R-strategists*), and 5 were classified as "*unclassified*", with 15 not included in Brun et al (2015)'s reproduction of Reynolds (2006). Given the paucity of *R-S-strategist* classifications (4 out 22 taxa with classifications) it would be inappropriate to specifically overlay the *R-S-strategist* framework on the taxonomic data we have collected.
- We have included discussion of the *R-S-strategist* classification in our re-worked discussion (from line 390).

**Additionally, species with large vs. small cells may show opposite responses to changing turbulence and wind regimes (Margalef, 1997, 1978).**

- We were not able to make any meaningful conclusions regarding cell size and shape and the degree of influence of the SAM (some species are always small, others are large and small)

**Finally, predicting the response of siliceous vs. calcareous taxa to SAM constitutes an exciting hypothesis: these groups have shown opposite responses to deeper mixing or nutrient entrainment (Cermeño et al., 2008).**

- The area studied was the Seasonal Ice Zone (SIZ) which is situated over the ocean upwelling zone of the Antarctic Divergence – nutrients in the surface waters of the SIZ are replenished over the six months when the sea surface is ice-covered and when there is almost no productivity (or consumption of nutrients). It is considered a high-nutrient, low-chlorophyll zone. In this region of annual winter nutrient replenishment, the influence of mixed-layer depth is less than most other areas of the world's ocean. The area falls outside the analysis of Cermeño et al., (2008), whose sampled area extended southward to only well north of the Antarctic Divergence, and would not be expected to conform to the trends observed by Cermeño et al., (2008) due to the replenishment of nutrients every winter in the area of our study.

**With respect to the clustering techniques used to describe communities I cannot give detailed recommendations, as the metrics used are beyond my expertise.**

**Detailed comments**

**There are too many comments to be listed. I therefore give examples for selected paragraphs, with comments on clarity, for each:**

**Abstract:**

**- Line 3: How many variables were tested?**

- We have amended the text to include

**- Line 6: How many species (genera, higher taxa) were included among the 22 taxa?**

- We have amended the text to include

**- Line 7: I do not understand 'CAP'. This term has not been introduced.**

- We have amended the text to include the full name of the CAP procedure

**- Lines 8, 9, 11, 17: The following terms are used: taxonomic community composition, taxa composition, phytoplankton community structure, taxonomic composition of phytoplankton. While I understand that the authors strive to include stylistic variation, the reader is confused by the multiple expressions. Do they denote the same thing or not? I recommend using use the same expression for the same thing. Else, once an expression is clear, an abbreviation of the latter may be used therein, as long as it denotes the same thing.**

- We have amended the text to make the terms consistent

**- Line 10: Unclear to me, if the correlation is significant or not.**

- We have amended the text to include "($p < 0.05$)" as appropriate

**- Line 13: Unclear to me, if "response" means a response of abundance or not.**

- We have amended the text to clarify: "relative-abundance response"

**- Line 15: Before, the expression "SAM index" was used, not "higher SAM". Does "higher SAM" refer to a more positive state of the SAM index?**

- We have amended the text for consistency in referring to the SAM

**- Line 17: Confusing, as taxonomic composition of phytoplankton is not the same thing as a standing stock (or a "pasture") of biomass of phytoplankton.**

- We have amended the text to clarify

**- Line 16: It is unclear to me, if the expression "pelagic ecosystem" is suitable in the context of a sea ice transition zone.**

- We have removed the word "*pelagic*" to avoid confusion

**- Line 16: It is unclear, how many of the total species that were studied, responded significantly to SAM. Thus, it is unclear, if this result is important or general.**

- We have amended the text to clarify

**- Line 10 ff: It is surprising that 'day of sampling' explains more variation in community composition than any other locally sampled environmental factor (SST, nutrients, etc). An interpretation on why this is the case would help the reader to assess the plausibility or importance of this result.**

- The seasonal ice zone has been previously observed and reported to have a winter period (around 6 months) with little or no phytoplankton productivity when the sea-surface is frozen, and a well-characterised bloom and systematic taxonomic succession through the spring-summer months as sea-ice melts – we will include this point in the Abstract (it is already in the Introduction and in the Discussion)

**Introduction:**

**- Line 21-23: The first two sentences are partially repetitive.**

- We have improved the text (line 26)

**- Line 21 ff: The paragraph wants to establish the importance of phytoplankton productivity in the study area for global phytoplankton productivity. While the reader understands that a larger fraction (~30%) of carbon fixed by phytoplankton is exported in the study region, relative to the global average (~ 20% exported) it remains unclear, if the study region is globally important. What is the area-weighted contribution of the study region to global phytoplankton C-export?**

- We have included a statement to this effect (line 33)

**1.2 The Southern Annular Mode**

**- Line 58 ff: Clarify the definition of SAM (see above). The reader cannot grasp how the sign of SAM is calculated or linked to changing pressure gradients, and thus how it is associated with physicochemical changes in the study system.**

- The calculation of SAM is beyond the scope of this paper, except in the most general terms – the SAM index used was calculated by NOAA (USA) and the wording of its description is as NOAA wants it reported.
- We have edited the text and added more simplistic description of the SAM: in the Introduction from line 66, and in the Methods line 160

**- Line 64 ff: SAM vs. SAM index vs. SAM state vs. SAM mode. Please use consistent expressions throughout the manuscript. …… and the use of "mode" in both the context of SAM and community composition may confuse the reader**

- We have removed all references to SAM maxima as MODES to remove confusion with MODE=MAXIMA and SOUTHERN ANNULAR MODE, that is, we have just defined the terms *SAM spring* and *SAM autumn* and refer to them by name without using the term "*mode*".

**In addition, "taxon" could always refer to both a species and a group of species.**

- We have modified the text to use "*taxa-group*" to refer to a single taxa-group

**2.1 Phytoplankton composition and abundance**

**- Line 116: One reads as if the abundance of phytoplankton communities was sampled. As much as I understand, the abundance of species or taxa was sampled. (Then, an abundance-weighted community composition was calculated?).**

- We have amended the text to remove any suggestion that "**abundance of phytoplankton communities was sampled**"

**2.3 Statistical analysis**

**- Line 151 ff: The methods section needs clarification, structurally and through editing. In this section, I have difficulties to understand whether three or more sets of analyses were performed based on the phytoplankton field data, and which of these analyses is most important to test the key hypothesis of the paper,**

- We will edit methods to improve clarity

**….. and at what temporal resolution the analyses were performed.**

- We have included more description of the derivation of the response surfaces to improve ease of understanding (from line 204)

**- Line 152: Has "community structure" really been correlated to "environmental covariates"? If I understand correctly, the abundance data was related to possible environmental drivers, per species. In this case, please specify: e.g. …and species abundance between samples**

- The correlation between the community structure (as determined from the ordination) and each environmental covariate was calculated according the procedure outlined in ter Braak (1995) and attributed to Dargie (1984) – we have included more explanation of CAP analysis for greater ease of interpretation for readers not familiar with CAP analysis (from line 185)

**- Line 151 ff: It is not clearly motivated, why clustering of community-level samples is suitable to identify the effect of SAM on community composition. To me, the number of 52 samples seems rather low already, and each degree of freedom may be valuable.**

- We have included more descriptive text to clarify (from Methods line 175, and Results line 276)

**3. Results**

**- The first results presented to the reader are abundance-distributions of taxa across samples. Yet, the reader might expect that the most important piece of evidence to elucidate the role of SAM for phytoplankton composition is first presented.**

- The logic behind the analysis and presentation of the data in relation to the hypothesis has been explained in the first paragraph of the *Results* section (line 220)
- We have re-arranged the *Results* section to put abundance information later, i.e. Section 3.1 "*Observed abundance*" has become Section 3.3 (line 313), with Section 3.1 becoming "*The influence of SAM on phytoplankton taxonomic composition*"

**- Line 206 ff. Can P, n, and R2-values be provided for the correlations?**

- The text has been modified to include specification of p with all in-text correlations
- P-values are not tabulated - currently reported correlations in tables are formatted to indicate when p<0.05 and when p<0.05/20 for Bonferroni correction

**- Table 1: I do not understand, why nutrients are excluded in this table.**

- nutrient levels are an effect of phytoplankton, not a cause, in this high-nutrient-low-chlorophyll environment that is nutrient-replenished through the winter – i.e. more growth through the productive spring-summer leads to less nutrition at the end of the summer – text has been included to explain this (line 157, 295, 425 onwards)

**- Figure 5. The caption remains vague. What are the "several underlying assumptions" of linear regression? Relevant to be discussed in the caption?**

- The caption has been improved by removal of the words "*though clearly several underlying assumptions of linear regression would not be met*" from the caption

**Overall, the manuscript requires a clear structure in order to show to what degree the SAM signals may matter to community composition, based on (ecological) hypotheses tested and data. The support in the data for this message, and the evaluation of the manuscript are complicated at current and warrant further attention.**

- We have made many changes throughout the manuscript to improve clarity and reduce wordage

[revised manuscript text omitted]

---

## Referee Report (RR1)

Second review for: Biogeosciences Discussions

Title: The Southern Annular Mode (SAM) influences phytoplankton communities in the seasonal ice zone of the Southern Ocean
Authors: Greaves et al.
Reviewer: Damiano Righetti, 12 March 2020

Summary of changes implemented:

A substantial effort has been made towards improvement of the ms. Several remaining points may further improve the quality of the ms and can be fixed. One corroborating statistical robustness test is encouraged.

Major remaining point:

It needs to become clear to the reader in the paper's discussion section whether the data shown support a *likely* or potential impact of SAM on phytoplankton community composition or whether they actually *demonstrate* such an effect. I think the former is the case, not the latter. The data at hand (i.e., 52 samples, spanning a relatively narrow geographic area of the total sea ice zone of Antarctica) and the methods used (correlational inference) cannot demonstrate any direct effect of SAM on phytoplankton composition in the Southern Ocean *yet*. To achieve this, future studies need to be based on wider data coverage and experimental incubations. On the other hand, I believe that the study establishes a valid hypothesis, i.e. that a SAM-phytoplankton linkage exists in the ice zone of the southern ocean, and this hypothesis is statistically supported by the (still rather limited) evidence for 12 out of 22 taxa.

The level of support or demonstration with regards to the main conclusions in the paper should be carefully re-checked, e.g.:
(1) Line 475 ff: "The present study demonstrates, for the first time, that variation in the SAM influences the community composition of phytoplankton in the SIZ (…)".
(2) Line 480ff: "We found that the Southern Annular Mode influenced phytoplankton community composition in the SIZ (…)". And, finally, in the same paragraph
(3), 485ff: "These observations suggest that the phytoplankton of the SIZ are indeed susceptible to changes in the SAM".
While the first two statements are rather optimistic on the basis of the relatively sparse data at hand (and hence might be clarified to: "This study demonstrates a statistical association between a large fraction of taxa and SAM… or " This study found statistical support/evidence for the influence of … Or ."SAM explained phytoplankton composition better than…"), the last statement appears very valid. In sum, I suggest being a bit more careful with any statements about what the study demonstrates. Note that the SAM itself is a proxy for several drivers that, in turn, influence the composition of phytoplankton. Also, SAM explained 13.3% of variation in composition (as stated in the abstract), while 86.7% remain unexplained.

I recommend checking/adjusting the strength of the conclusion made in line 303ff to the strength of conclusion made with regards to community composition statements (examples above), as exactly the same number of samples (data power) was used there (but measuring chlorophyll).

Minor points of critique and suggestions:

- The authors mention several times that it is the first study to show an influence of SAM on phytoplankton composition in the SIZ of the SO. This does not need to be stated more than once. The value of the study is emerging from a careful presentation and interpretation of results, rather than the claim of novelty (and the study's novelty, ideally, already emerges from the literature review in the introduction).

- I think a remarkable piece of evidence is that the study finds a statistical association between community composition and SAM, but only for "SAM August" and "SAM Spring", unlike not for "SAM winter" (when there is sea ice cover). This point may deserve more space in the paper's discussion section. Second, if I understood correctly, a similar pattern emerges from the independent, remotely sensed Chl data, providing additional support to the ineffectiveness of "SAM winter".

- I have criticized during the first Review that there was a lack of clarity in several instances, and that the argumentation lines or paragraph structures were partially broken. While it is the task of the authors to make sure that these points are fixed, I provide a few more inputs herein that were apparent during the revisit of the ms:

- Clarity of SAM definition line 71 ff: i.e.: "The SAM is estimated either from station measurements as the difference in normalized zonal mean atmospheric sea-level pressure between 40∘ S and 65∘ S (Gong and Wang, 1999; Marshall, 2003), or from Principal Component analysis of gridded data of atmospheric pressure or temperature, at sea-level or at a geopotential height (Ho et al., 2012)."
For the reader, it is
(a) unclear if air pressure at 40° S is subtracted from the air pressure at 65°S - or vice versa (or if the absolute difference is taken) - and
(b) unclear, if the Principal Component method involves a comparison of the same latitudinal positions or not. Please clarify.

-  Clarity of community composition definition: Phytoplankton community composition (as used currently in the paper) both denotes the number of taxa found and their relative cell counts/contribution to the total cell counts in a sample. Hence, species identities and their abundance are both included in the definition. I hence, advise against using "taxonomic composition" in line 233 or "community taxonomic composition" in line 241 as synonyms, as they mean something else (e.g., contribution of (larger) taxa to the community). Please be consistent in use of terms.

- I acknowledge the authors' effort to split the 22 taxa investigated under the microscopes into test groups. Obviously the information available was insufficient to perform such a test. I hence, do not insist to include lines 391-395 in the ms. (Yet, I agree that it is a noteworthy point that a significant fraction of the taxa in the study are endemic to the region and/or lack observation records in OBIS from other regions).

- I proposed to put primary productivity in the SIZ into the context of global primary productivity to give the reader a sense of the importance of the study region. This has not been implemented. The sentence now reads as follows, line 34 ff: "Total productivity within the SIZ of the SO has been estimated at 68–107 Tg C yr$^{-1}$ from 1997 to 2005 (Arrigo et al., 2008), corresponding to roughly one third/fourth… of the

…. Tg C yr produced globally, and consequently SO phytoplankton play a role in mitigating the accumulation of anthropogenic greenhouse gases in the word's atmosphere (…).
Could the authors include a statement (see my red textual edits) to provide global context? Only this would enable the reader to grasp the global relevance of the SIZ.

- Caption of Figure 2, second line. Taxa codes listed in Table 3 not Table 2, I guess.

- CAP method explanation: It would be helpful – at an appropriate position in the paper –to briefly explain to the reader how community composition was related to environmental covariates in CAP. The reader does not intuitively comprehend, if (a) community composition is reduced to a single expression and then related to the environmental covariate, or if (b) the abundance values of each species are related to the covariate, simultaneously, in such CAP analysis. Please clarify.

- Comment: I wondered why the clustering of community samples was used as a strategy to test the hypothesis of SAM influence on composition. If I understand now correctly, it served to provide additional context to the data shown in Figure 6 and hence served as a complementary analysis.

- Results section: Line 223: This sentence interprets the results. In the results section, I suggest to present the (main) results first, without interpretation. Also slightly confusing: Cluster analysis is introduced here as a method.

- Statistics: I acknowledge that the authors present $p$ values in many instances. In Figure 5, I suggest to present $r$-squared values inside each panel, rather than $r$ values, as $r$-squared values reflect the variance explained, which is often referred to elsewhere in the paper. (Please also stick to "variance", or "variation", in the ms).

- Replicates per sample: Does figure 5 show sample means (of the three replicates per sample)? If so, could standard deviations be added to each dot to depict how much variability was involved between replicates per sample? The reader gets no sense at the moment about the within-sample variability/uncertainty between replicates.

- Line 41ff: Indeed, it has been estimated that productivity declines by 1%. Yet the paper has been disputed. I hence suggest using e.g. past tense, not present, and to refer to a criticizing paper (e.g. brief communication by David Mackas: DOI 10.1038/nature09951) as well.

- Line 21vs line 61. Contradictory.

- Line 153: Has a parallel analysis using absolute abundances instead of relative abundances, been attempted as a robustness test? (I.e., How many taxa do show positive associations with SAM in this case?). I strongly recommend running such a parallel test in order to demonstrate the statistical robustness/sensitivity of results.

- Line 221: Figure 3. Could the same x-axis scales be used in both panels? Slightly confusing why panel a) uses months and days while panel b) uses days only.

- Line 224: These (plural) vs. other analysis (singular). Mismatch.

- Line 265ff: The sentence is not clear to me, "most variation due to seasonal succession due to". Perhaps: Most variation *in* the seasonal succession of.. ?

- Line 278ff: I suggest to replace SIMPROF by "similarity profile analysis" or similar, as the reader does not remember what SIMPROF stands for.

- Line 281ff: This is a methodological argument, could this be introduced earlier on? E.g. in the methods section.

- Line 296ff: I do not think that indicators were derived. I argue that relationships were examined or tested by the data. I agree, however, that this methodological statement may help the reader (else it should be omitted in the results section).

- Line 314: I suggest deleting "taxonomic". It is not the abundance of taxa/species, but the abundance of cells per species or taxon-group that matters.

- Line 324ff: I do not fully comprehend this point. Why should a demonstration of separation of samples into clusters support the conclusion that SAM affects community composition (stated in line 322)?

- Line 356ff: Please specify: Was a statistical association between SAM indices and phytoplankton composition found (?).

- Line 372: Comment: Why not including this expectation (SAM winter has no effect vs. other SAM have effect) in the main hypothesis stated upfront in the paper? If this expectation is stated here, a chance to incept this fascinating idea earlier on to the reader is being missed. The conclusion in line 384 (potentially adjust, see main comment above) may then actually be expected (adjust if needed).

- Line 388ff: This sentence is rather long. I do not get the key point / essence of it.

- Line 404ff paragraph: What does this paragraph mean/suggest? Can a final sentence or statement summarize the message to the reader?

- Line 432: "that" is missing in this sentence.

- Line 435ff: The first phrase (i.e., "The maxima in the variance in total chlorophyll explained by the SAM …" is hard to understand. (Rephrasing possible?).

- 4.4. Implications. Line 446ff This first paragraph provides an excellent embedment of the study's result in the literature. However, this seems to be a key aspect of the discussion, rather than an implication of the study. Also the paragraph in line 463ff seems to be rather a discussion point than an implication.

- Line 463: "It is not surprising that climate in both autumn and spring influence (…)" Climate or weather mode? The sentence treats the hypothesis that is established in the paper as granted. Please see my point above on the degree of the study's conclusion.

- Line 468ff: "The surprise is that (…)". Wasn't this expected based on the hypothesis of the paper?

---

## Author Response (AR2)

Response to comments on the submitted manuscript: Greaves et al. - *The Southern Annular Mode (SAM) influences phytoplankton communities in the seasonal ice zone of the Southern Ocean*

29 April 2020
* * *
We thank the reviewer for their valuable feedback on this manuscript. These have identified several areas for improvement of the manuscript, which we have addressed below:

**RC2 - Damiano Righetti**

Second review for: Biogeosciences Discussions

Title: The Southern Annular Mode (SAM) influences phytoplankton communities in the seasonal ice zone of the Southern Ocean

Authors: Greaves et al.

Reviewer: Damiano Righetti, 12 March 2020

Summary of changes implemented:

A substantial effort has been made towards improvement of the ms. Several remaining points may further improve the quality of the ms and can be fixed. One corroborating statistical robustness test is encouraged.

Major remaining point:

It needs to become clear to the reader in the paper's discussion section whether the data shown support a likely or potential impact of SAM on phytoplankton community composition or whether they actually demonstrate such an effect. I think the former is the case, not the latter. The data at hand (i.e., 52 samples, spanning a relatively narrow geographic area of the total sea ice zone of Antarctica) and the methods used (correlational inference) cannot demonstrate any direct effect of SAM on phytoplankton composition in the Southern Ocean yet. To achieve this, future studies need to be based on wider data coverage and experimental incubations. On the other hand, I believe that the study establishes a valid hypothesis, i.e. that a SAM-phytoplankton linkage exists in the ice zone of the southern ocean, and this hypothesis is statistically supported by the (still rather limited) evidence for 12 out of 22 taxa.

**The level of support or demonstration with regards to the main conclusions in the paper should be carefully re-checked, e.g.:**

**(1) Line 475 ff: "The present study demonstrates, for the first time, that variation in the SAM influences the community composition of phytoplankton in the SIZ (…)". (2) Line 480ff: "We found that the Southern Annular Mode influenced phytoplankton community composition in the SIZ (…)". And, finally, in the same paragraph (3), 485ff: "These observations suggest that the phytoplankton of the SIZ are indeed susceptible to changes in the SAM". While the first two statements are rather optimistic on the basis of the relatively sparse data at hand (and hence might be clarified to: "This study demonstrates a statistical association between a large fraction of taxa and SAM… or " This study found statistical support/evidence for the influence of … Or ."SAM**

explained phytoplankton composition better than…"), the last statement appears very valid. In sum, I suggest being a bit more careful with any statements about what the study demonstrates. Note that the SAM itself is a proxy for several drivers that, in turn, influence the composition of phytoplankton. Also, SAM explained 13.3% of variation in composition (as stated in the abstract), while 86.7% remain unexplained.

- We have recrafted the conclusion to be more specific to the results
- We have modified instances where we claimed "*SAM was influencing phytoplankton*" or similar to "*variation in SAM is related to variation in phytoplankton*" or similar
- "**86.7% remain unexplained**" – this is incorrect: 62.5% of variation in composition is unexplained (Table 1b)

I recommend checking/adjusting the strength of the conclusion made in line 303ff to the strength of conclusion made with regards to community composition statements (examples above), as exactly the same number of samples (data power) was used there (but measuring chlorophyll).

- We have recrafted the supporting conclusion made here in respect of chlorophyll to be more speculative

Minor points of critique and suggestions:

- The authors mention several times that it is the first study to show an influence of SAM on phytoplankton composition in the SIZ of the SO. This does not need to be stated more than once. The value of the study is emerging from a careful presentation and interpretation of results, rather than the claim of novelty (and the study's novelty, ideally, already emerges from the literature review in the introduction).

- We have removed all but one reference to this being the first study to have observed the relationship between SAM and phytoplankton community composition

- I think a remarkable piece of evidence is that the study finds a statistical association between community composition and SAM, but only for "SAM August" and "SAM Spring", unlike not for "SAM winter" (when there is sea ice cover). This point may deserve more space in the paper's discussion section. Second, if I understood correctly, a similar pattern emerges from the independent, remotely sensed Chl data, providing additional support to the ineffectiveness of "SAM winter".

- We have included mention of SAM winter in the discussion

- I have criticized during the first Review that there was a lack of clarity in several instances, and that the argumentation lines or paragraph structures were partially broken. While it is the task of the authors to make sure that these points are fixed, I provide a few more inputs herein that were apparent during the revisit of the ms:

- Clarity of SAM definition line 71 ff: i.e.: "The SAM is estimated either from station measurements as the difference in normalized zonal mean atmospheric sea-level pressure between 40° S and 65° S (Gong and Wang, 1999; Marshall, 2003), or from Principal Component analysis of gridded data of atmospheric pressure or temperature, at sea-level or at a geopotential height (Ho et al., 2012)."

For the reader, it is

(a) unclear if air pressure at 40° S is subtracted from the air pressure at 65°S - or vice versa (or if the absolute difference is taken) - and

**(b) unclear, if the Principal Component method involves a comparison of the same latitudinal positions or not. Please clarify.**

- Re: (a). Neither is used here - we use NOAA's daily product which calculates the projection of the 700 hPa GPH anomaly onto the loading pattern of the first EOF in the southern hemisphere. However we have clarified to say that the difference is "40 minus 65" Ref: Gong and Wang, 1999
- Re (b). We apologise the confusing way this was written! While we outlined the two common methods of calculating SAM, we neglected to highlight which approach we chose. We edited the text in two places to resolve this.
- Edits at around line 58 and line 140

**- Clarity of community composition definition: Phytoplankton community composition (as used currently in the paper) both denotes the number of taxa found and their relative cell counts/contribution to the total cell counts in a sample. Hence, species identities and their abundance are both included in the definition. I hence, advise against using "taxonomic composition" in line 233 or "community taxonomic composition" in line 241 as synonyms, as they mean something else (e.g., contribution of (larger) taxa to the community). Please be consistent in use of terms.**

- All instances of **"community taxonomic composition"** and **"taxonomic composition"** have been removed in favour of **"phytoplankton community composition"**

**- I acknowledge the authors' effort to split the 22 taxa investigated under the microscopes into test groups. Obviously the information available was insufficient to perform such a test. I hence, do not insist to include lines 391-395 in the ms. (Yet, I agree that it is a noteworthy point that a significant fraction of the taxa in the study are endemic to the region and/or lack observation records in OBIS from other regions).**

- Some of the text has been removed, the reference to the number of taxa represented in OBIS has been retained.

**- I proposed to put primary productivity in the SIZ into the context of global primary productivity to give the reader a sense of the importance of the study region. This has not been implemented. The sentence now reads as follows, line 34 ff: "Total productivity within the SIZ of the SO has been estimated at 68–107 Tg C yr-1 from 1997 to 2005 (Arrigo et al., 2008), corresponding to roughly one third/fourth… of the …. Tg C yr produced globally, and consequently SO phytoplankton play a role in mitigating the accumulation of anthropogenic greenhouse gases in the word's atmosphere (…). Could the authors include a statement (see my red textual edits) to provide global context? Only this would enable the reader to grasp the global relevance of the SIZ.**

- The requested information has been added to the manuscript

**- Caption of Figure 2, second line. Taxa codes listed in Table 3 not Table 2, I guess.**

- The error in Figure 2 caption has been corrected

**- CAP method explanation: It would be helpful – at an appropriate position in the paper –to briefly explain to the reader how community composition was related to environmental covariates in CAP. The reader does not intuitively comprehend, if (a) community composition is reduced to a single expression and then related to the environmental covariate, or if (b) the abundance values of each species are related to the covariate, simultaneously, in such CAP analysis. Please clarify.**

- The text has been modified address this point

**- Comment: I wondered why the clustering of community samples was used as a strategy to test the hypothesis of SAM influence on composition. If I understand now correctly, it served to provide additional context to the data shown in Figure 6 and hence served as a complementary analysis.**

- The text has been modified address this point

**- Results section: Line 223: This sentence interprets the results. In the results section, I suggest to present the (main) results first, without interpretation. Also slightly confusing: Cluster analysis is introduced here as a method.**

- The text has been modified address this point

**- Statistics: I acknowledge that the authors present p values in many instances. In Figure 5, I suggest to present r-squared values inside each panel, rather than r values, as r-squared values reflect the variance explained, which is often referred to elsewhere in the paper.**

- r2 has now replaced r in Figure 5

**(Please also stick to "variance", or "variation", in the ms).**

- Some instances of the word "*variation*" have been changed to "*variance*" to be consistent

**- Replicates per sample: Does figure 5 show sample means (of the three replicates per sample)? If so, could standard deviations be added to each dot to depict how much variability was involved between replicates per sample? The reader gets no sense at the moment about the within-sample variability/uncertainty between replicates.**

- The individual "*SEM fields*" … "*assessed for each sample*" (line 134) were not considered to be sampling replicates, but rather incremental increases in the area of each sample analysed, and thus statistical confidence limits cannot be calculated – we have added the words: "*– these were considered as incremental increases in the sample area covered and not sampling replicates*"

**- Line 41ff: Indeed, it has been estimated that productivity declines by 1%. Yet the paper has been disputed. I hence suggest using e.g. past tense, not present, and to refer to a criticizing paper (e.g. brief communication by David Mackas: DOI 10.1038/nature09951) as well.**

- The text has been amended and further information cited

**- Line 21vs line 61. Contradictory.**

- They are not contradictory – Line 21 is referring to "*extreme seasonal variability*" and line 61 is referring to "*phytoplankton bloom development*" – whilst we consider this to be clear we have switched "*the*" to "*a*" in the sentence that was line 61

**- Line 153: Has a parallel analysis using absolute abundances instead of relative abundances, been attempted as a robustness test? (I.e., How many taxa do show positive associations with SAM in this case?). I strongly recommend running such a parallel test in order to demonstrate the statistical robustness/sensitivity of results.**

- Parallel analyses undertaken with absolute abundances showed similar but slightly weaker results. It has now been reported in Supplementary Material, and a paragraph summarising the similarities has been added to the RESULTS section

**- Line 221: Figure 3. Could the same x-axis scales be used in both panels? Slightly confusing why panel a) uses months and days while panel b) uses days only.**

- We would prefer to keep the axis as they are, being different – because they represent quite different ways of expressing the timing of the SAM "window". The seasonal same maxima are more easily interpreted with a month scale rather than with a days-in-the-year scale, and the time-before-sampling SAM prior maxima has no monthly meaning as the samples are collected over a 3-month window.

**- Line 224: These (plural) vs. other analysis (singular). Mismatch.**

- This has been corrected.

**- Line 265ff: The sentence is not clear to me, "most variation due to seasonal succession due to". Perhaps: Most variation in the seasonal succession of.. ?**

- We have reworded the sentence to improve clarity

**- Line 278ff: I suggest to replace SIMPROF by "similarity profile analysis" or similar, as the reader does not remember what SIMPROF stands for.**

- "*SIMPROF*" has been replaced with "*similarity profile (SIMPROF) permutation analysis*" in the sentence

**- Line 281ff: This is a methodological argument, could this be introduced earlier on? E.g. in the methods section.**

- We have moved this explanation to the methods section

**- Line 296ff: I do not think that indicators were derived. I argue that relationships were examined or tested by the data. I agree, however, that this methodological statement may help the reader (else it should be omitted in the results section).**

- We have replaced the word "*derived*" with the word "*obtained*"

**- Line 314: I suggest. It is not the abundance of taxa/species, but the abundance of cells per species or taxon-group that matters.**

- deleted "*taxonomic*" from section heading

**- Line 324ff: I do not fully comprehend this point. Why should a demonstration of separation of samples into clusters support the conclusion that SAM affects community composition (stated in line 322)?**

- We have added more explanation "……*if clustering had revealed few or no clusters it would have been indicative of levels of community variance (either high or low) unlikely to be systematically explainable with the environmental variables.*"

**- Line 356ff: Please specify: Was a statistical association between SAM indices and phytoplankton composition found (?).**

- We have added the statistics to paragraph

**- Line 372: Comment: Why not including this expectation (SAM winter has no effect vs. other SAM have effect) in the main hypothesis stated upfront in the paper? If this expectation is stated here, a chance to incept this fascinating idea earlier on to the reader is being missed. The conclusion in line 384 (potentially adjust, see main comment above) may then actually be expected (adjust if needed).**

- We have modified the text

**- Line 388ff: This sentence is rather long. I do not get the key point / essence of it.**

- We have removed the paragraph as the reviewer did not grasp the point we were making and it adds little to the story

**- Line 404ff paragraph: What does this paragraph mean/suggest? Can a final sentence or statement summarize the message to the reader?**

- We have removed this paragraph as not contributing to the main theme of the manuscript

**- Line 432: "that" is missing in this sentence.**

- "that" added

**- Line 435ff: The first phrase (i.e., "The maxima in the variance in total chlorophyll explained by the SAM …" is hard to understand. (Rephrasing possible?).**

- We have modified the sentence to improve clarity

**- 4.4. Implications. Line 446ff This first paragraph provides an excellent embedment of the study's result in the literature. However, this seems to be a key aspect of the discussion, rather than an implication of the study. Also the paragraph in line 463ff seems to be rather a discussion point than an implication.**

- We have moved and edited these discussion paragraphs out of the "*Implications*" section and into the "*Discussion*" section

**- Line 463: "It is not surprising that climate in both autumn and spring influence (…)" Climate or weather mode? The sentence treats the hypothesis that is established in the paper as granted. Please see my point above on the degree of the study's conclusion. - Line 468ff: "The surprise is that (…)". Wasn't this expected based on the hypothesis of the paper?**

- We have taken reference to "*surprise*" out of the paragraph

[revised manuscript text omitted]